# Mechanical forces and ligand binding modulate *Pseudomonas aeruginosa* PilY1 mechanosensitive protein

Francisco J Cao-Garcia[1,2] , Jane E Walker[3], Stephanie Board[3], Alvaro Alonso-Caballero[3]

**Surface sensing initiates bacterial colonization of substrates. The protein PilY1 plays key roles during this process—surface detection, host adhesion, and motility—while experiencing mechanical perturbations of varying magnitudes. In *Pseudomonas aeruginosa*, the adhesion and motility functions of PilY1 are associated with integrin and calcium ligand-binding sites; however, how mechanical forces influence PilY1's dynamics and its interactions with these ligands remain unknown. Here, using single-molecule magnetic tweezers, we reveal that PilY1 is a mechanosensor protein that exhibits different behaviors depending on the force load. At high forces (>20 pN), PilY1 unfolds through a hierarchical sequence of intermediates, whose mechanical stability increases with calcium binding. This enhanced stability may help counteract type IV pilus retraction forces during motility. At low forces (<7 pN), we identify the dynamics of the integrin-binding domain, which is reminiscent of the behavior of mechanosensor proteins. Integrin binding induces a force-dependent conformational change in this domain, shortening its unfolded extension. Our findings suggest that PilY1 roles are force- and ligand-modulated, which could entail a mechanical-based compartmentalization of its functions.**

## Introduction

Mechanical forces play key roles in cell fate, triggering crucial events such as differentiation, proliferation, or motility (Eyckmans et al, 2011). Mechanical sensing and the responses elicited by force are well-known phenomena in eukaryotic cells; however, our understanding of how prokaryotes sense and respond to mechanical cues is still scarce (Gordon & Wang, 2019).

Although explored to a lower extent, it is recognized that mechanical stimuli promote critical life changes in prokaryotes (Persat et al, 2015; Chawla et al, 2020; Dufrêne & Persat, 2020). Concretely, the mechanical cues induced by surface proximity stimulate the switch from a planktonic to a sessile lifestyle (Persat, 2017). In pathogenic bacteria like *Pseudomonas aeruginosa*, surface detection is followed by colonization, biofilm formation, and virulence development toward a host (Mordue et al, 2021). It has been proposed that the protein PilY1 may act as a mechanosensor mediating surface detection (Rodesney et al, 2017). This protein is located at the tip of the type IV pili (Treuner-Lange et al, 2020) (T4P), the motor-driven appendages that power bacterial translocation on substrates (O'Toole & Kolter, 1998; Persat et al, 2015; Craig et al, 2019). In addition to surface sensing, PilY1 is involved in T4P biogenesis and participates in several of its functions (Nguyen et al, 2015), such as in twitching motility (Orans et al, 2010), adhesion (Hoppe et al, 2017), biofilm formation, and virulence (Siryaporn et al, 2014; Marko et al, 2018). Therefore, PilY1 stands out as a multipurpose protein with a pivotal role in numerous processes that govern the surface-associated life of bacteria (Kuchma et al, 2010).

The different roles of PilY1 have been pinned down to specific features found in its sequence, which has enabled the compartmentalization of the protein into two functional sections (Fig 1A). On the N terminus, the von Willebrand A (vWA) domain has been assigned the mechanosensing role of PilY1. This function was inferred from the phenotype manifested after vWA domain deletion, which placed planktonic bacteria in a constitutive virulence state, a behavior typically observed after surface engagement (Siryaporn et al, 2014). The absence of the vWA domain mimicked the signaling triggered after PilY1 interaction with a surface, suggesting a role in mechanotransduction. More recently, it was found that the vWA domain could also have an adhesive function linked to conserved cysteine residues of its sequence. Mutation of these residues reduced the adhesion strength of bacteria and dwindled the responses characteristic of surface-committed cells (Webster et al, 2021). Although genetic, surface mechanics (Siryaporn et al, 2014; Wang et al, 2023), and biophysical (Webster et al, 2021) studies support the vWA domain's involvement in mechanosensing, less is known about how force could regulate the functions of the C-terminal section of PilY1.

Unlike the vWA domain, the sequence of the C-terminal part of PilY1 is well conserved across species, and its structure has been

---

[1]Departamento de Estructura de la Materia, Física Térmica y Electrónica, Universidad Complutense de Madrid, Madrid, Spain    [2]IMDEA Nanociencia, Madrid, Spain
[3]Department of Physics, King´s College London, London, UK

Correspondence: alvaro.alonsoc@uam.es
Stephanie Board's present address is Laboratoire des Biomolécules, École Normale Supérieure, Université PSL, CNRS, Sorbonne Université, Paris, France
Alvaro Alonso-Caballero's present address is Departamento de Biología, Universidad Autónoma de Madrid, Madrid, Spain

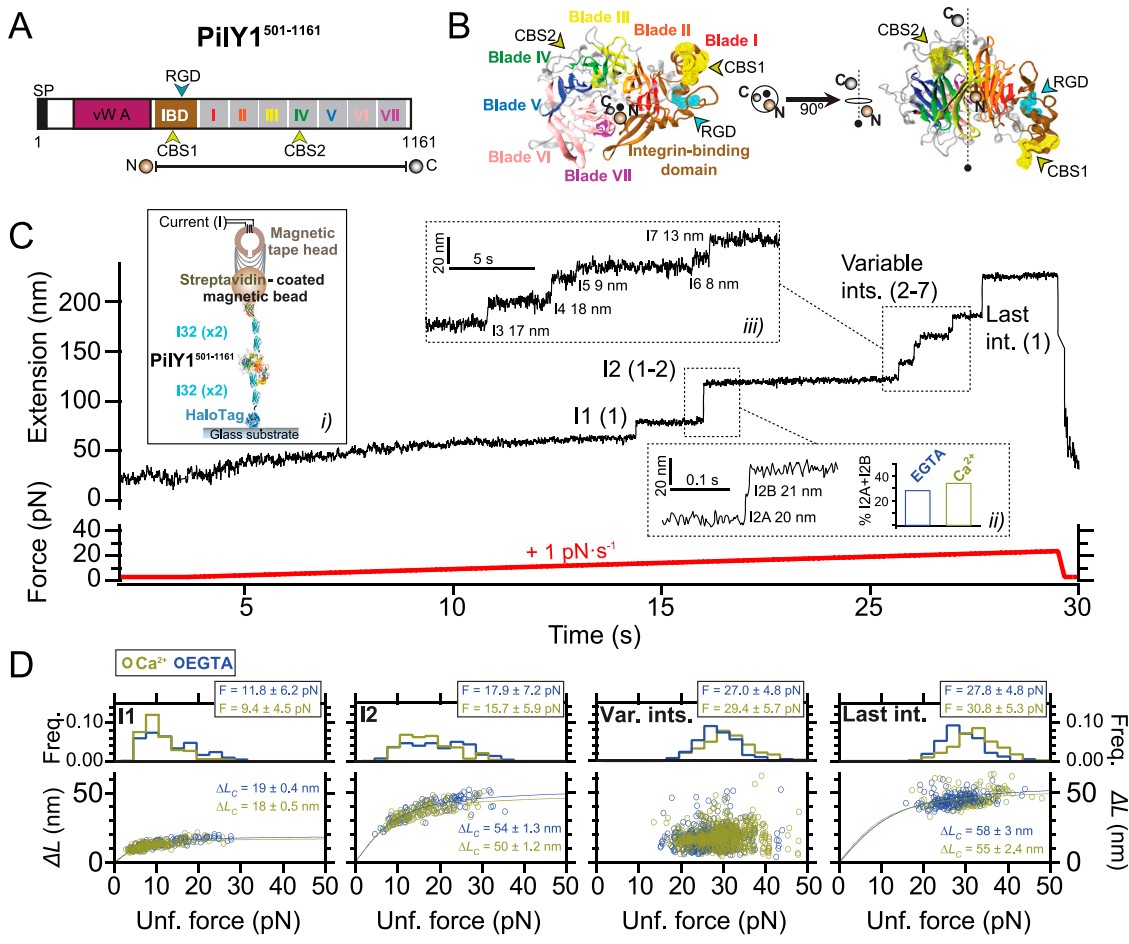

**Figure 1. PilY1$^{501-1161}$ mechanical hierarchy and Ca$^{2+}$ dual effect.**
**(A)** Scheme of the PilY1 protein sequence and structural features (SP: signal peptide; vWA domain: von Willebrand A domain; IBD: integrin-binding domain; RGD: integrin-binding site; CBS1: Ca$^{2+}$-binding site 1; blades I to VII of the $\beta$-propeller; CBS2: Ca$^{2+}$-binding site 2). The sequence employed spans from residues 501–1,161. **(B)** Crystal structure of PilY1$^{501-1161}$ from top (left) and side (right) views (prediction from AlphaFold [Jumper et al, 2021; Varadi et al, 2022] and color assignation based on reference [Orans et al, 2010]). **(C)** Magnetic tweezer force-ramp trajectory of PilY1$^{501-1161}$ (inset *i*; see the main text and methods for a full description of the chimeric protein construct and tethering) under a 1-pN·s$^{-1}$ loading rate. PilY1$^{501-1161}$ unfolds through multiple intermediates, which are classified based on their order of appearance: I1 (1 event), I2 (1 or 2 consecutive events, inset *ii*), variable intermediates (two to seven events of different sizes, inset *iii*), and last intermediate (1 event). **(D)** Unfolding force distributions (top, mean ± SD) and force dependency of unfolding extensions (bottom) per intermediate group and condition (Ca$^{2+}$ or EGTA). The extension ($\Delta L$) versus unfolding force dependency is described with the freely jointed chain model (FJC) for polymer elasticity (lines in the bottom panels are fits of this model). The Kuhn length values, $l_K$, in EGTA for I1, I2, and the last intermediate are 1.5 ± 0.1 nm, 1.0 ± 0.1 nm, and 0.7 ± 0.2 nm, and in Ca2+ are 1.7 ± 0.1 nm, 1.1 ± 0.1 nm, and 0.9 ± 0.2 nm.

experimentally determined (Orans et al, 2010). In *P. aeruginosa* strains, the sequence spanning residues ~640–1,150 adopts a 7-bladed $\beta$-propeller configuration. This type of fold comprises several structural repeats, termed blades, arranged around a central axis (Orans et al, 2010; Mylemans et al, 2021) (Fig 1A and B). Blade IV contains an EF-hand Ca$^{2+}$-binding motif that modulates the extension and retraction cycles of the T4P. In the metal-bound state, PilY1 inhibits the retraction activity of the T4P machinery, which can exert pulling forces ranging 30–100 pN (Maier et al, 2002; Ribbe et al, 2017). After Ca$^{2+}$ release, pilus retraction is restored, indicating that the PilY1 C-terminal domain acts as a Ca$^{2+}$-dependent switch that modulates T4P dynamics in biogenesis and twitching motility (Orans et al, 2010). Further scrutiny of the PilY1 sequence identified an adhesin function between the vWA domain

and $\beta$-propeller. PilY1 proteins from *P. aeruginosa* strains harbor an RGD motif that binds to the extracellular domains of integrin. A second Ca$^{2+}$-binding site was found in the vicinity of the RGD motif, and it was demonstrated that integrin binding required both the RGD motif and Ca$^{2+}$ binding on this site (Johnson et al, 2011). Integrins are transmembrane proteins present in the respiratory epithelium, a common niche of *P. aeruginosa* pathogenic strains (Heiniger et al, 2010; Jurado-Martín et al, 2021).

PilY1 plays a key role in surface-related processes and is exposed to mechanical perturbations. Understanding the force response and ligand-binding modulation of PilY1 could shed light on the molecular mechanisms underlying surface colonization. Herein, we have employed a single-molecule approach to address the nanomechanics of the PilY1 C-terminal section, which holds the

ligand-binding sites responsible for T4P dynamics and host cell adhesion. Our results indicate that PilY1 is a mechanosensor protein that exhibits different behaviors depending on the force load. In the high-force regime, PilY1 unfolds through multiple intermediates in a complex yet hierarchical sequence affected by $Ca^{2+}$. When this metal is bound to its furthermost C-terminal site, which regulates T4P dynamics, PilY1 stability increases and becomes less sensitive to mechanical perturbations. In the low-force regime, the integrin-binding domain undergoes folding and unfolding transitions in a 2 pN range, showing a steep force dependency characteristic of mechanosensitive proteins. Experiments with integrin enabled us to detect single binding events, which partially blocked the unfolding of its cognate domain and induced a shortening of the protein. We conclude that the PilY1 C-terminal section exhibits a dual force dependency where its functions as an adhesin and T4P regulator could be invoked depending on the mechanical load experienced by bacteria during substrate colonization. Our work establishes a mechanical-based connection between the structure and functions of PilY1 and suggests that this bacterial protein has a force-sensing activity that extends beyond its N-terminal domain. These findings underpin the intricate and functional wealth of PilY1.

## Results

### Hierarchical unfolding and $Ca^{2+}$-dependent mechanical stability

To explore the nanomechanics of PilY1, we employed single-molecule magnetic tweezers (Gosse & Croquette, 2002). This technique allows the application of well-calibrated forces to single proteins and monitoring their conformational changes with high spatial, temporal, and force resolutions (Popa et al, 2016; Zhao et al, 2017). To conduct single protein measurements, we designed a chimeric polyprotein with features that enable its end-to-end specific tethering between a functionalized glass cover slide and a superparamagnetic bead, the force probe. Our protein of interest is the C-terminal section of PilY1 encompassing residues 501–1,161 from the *P. aeruginosa* PAO1 strain (PilY1$^{501-1161}$) (Fig 1A and B). In the chimeric construct (Fig 1C, inset *i*), the N terminus presents a HaloTag protein for covalent anchoring on the glass surface (Taniguchi & Kawakami, 2010), and the PilY1$^{501-1161}$ sequence is flanked on both sides by two copies of the human titin I32 domain, which provide spacing between the surface and the bead. HaloTag and I32 domains exhibit high mechanical stability and act as rigid molecular handles (Li et al, 2002; Popa et al, 2013). At the forces tested in this study, their probability of unfolding is low, and they do not exhibit conformational changes during the experiments (see the Materials and Methods section). The C terminus of the polyprotein contains a biotinylated AviTag sequence that allows the tethering to a streptavidin-coated superparamagnetic bead.

Our first goal was to investigate the mechanical stability and fingerprint of PilY1$^{501-1161}$ under a steadily increasing force load. We also aimed to assess whether $Ca^{2+}$ impacts its stability, as reported in protein mechanics (Echelman et al, 2016; Scholl et al, 2016; Milles et al, 2018). For this reason, we conducted experiments with $Ca^{2+}$ or

the calcium chelator EGTA. Fig 1C shows an unfolding trajectory of this construct, which was exposed to a loading rate of 1 pN•s$^{-1}$ from its folded state up to its complete unfolding. As the load increases, the protein is stretched, and discrete jumps in the extension can be observed corresponding to the unfolding of PilY1$^{501-1161}$ through multiple intermediates. These unfolding events are heterogeneous in extension and number; nevertheless, we noticed a conserved sequence. We assigned a number to each intermediate based on their order of appearance. In this trajectory, the least stable intermediate, I1, unfolds in a single 14-nm event at 13 pN. At 15 pN, we detect the unfolding of I2, which usually occurs as a single event, but, as shown in Fig 1C, inset *ii*, it can proceed through two sub-intermediates. After I1 and I2, a population of unfolding events of heterogeneous quantity and size appears (Fig 1C, inset *iii*). The number of intermediates oscillates inter- and intramolecularly (between unfolding pulses) and ranges from two large events to up to seven smaller extensions. Because of the difficulty of identifying reproducible conformations, we pooled them into a single group termed variable intermediates. The final ~40 nm event in this trajectory, occurring at 22 pN, was named the last intermediate. Therefore, the unfolding pattern of PilY1$^{501-1161}$ displays a clear fingerprint of four classes of events occurring hierarchically, with a conserved sequence of I1 (1 event), I2 (1–2 events), variable intermediates (2–7 events), and last intermediate (1 event).

In Fig 1D, we show the unfolding force distribution (top) and the force-dependent step size (bottom) of each of the classes of intermediates detected in the presence of $Ca^{2+}$ or EGTA. By collecting data from multiple trajectories and molecules, we obtain the scatter plots shown in Fig 1D, which depict single unfolding events of each class occurring over a distribution of forces. The step size ($\Delta L$) force dependency of each class can be described by polymer models such as the freely jointed chain (FJC) (Flory, 1956). From the fits of the data to this model, we can obtain the extension of the unfolded intermediates ($\Delta L_c$). In the case of I1, I2, and the last intermediate classes, the data points are well described by a single fit in both $Ca^{2+}$ and EGTA, indicating that these events can be attributed to unique structural features in PilY1$^{501-1161}$. In ~30% of the trajectories, the I2 intermediate proceeds through two consecutive sub-intermediates (Fig 1C, inset *ii*, and Fig S1A), which can be described with two separate fits whose combined $\Delta L_c$ matches the $\Delta L_c$ of I2 when detected as a single event (Fig S1B). In the variable intermediate class, the unfolding events cannot be described with an FJC fit because of its heterogeneity. Nevertheless, insights can be obtained from the unfolding force means of each group and the classification standard employed (Fig 1D, top graphs). $Ca^{2+}$ decreases the average unfolding force of both I1 and I2 (11.8 ± 0.5 pN to 9.4 ± 0.3 pN, for I1; 17.9 ± 0.7 pN to 15.7 ± 0.6 pN, for I2; mean ± SEM; see the Materials and Methods section and Table S1), also observed when I2 occurs in two events (Fig S1B). In contrast, $Ca^{2+}$ increases the collective stability of the variable intermediates (27.0 ± 0.2 pN to 29.4 ± 0.2 pN) and the last intermediate (27.8 ± 0.4 pN to 30.8 ± 0.4 pN). $Ca^{2+}$ has opposite effects on the PilY1$^{501-1161}$ structure, lowering the mechanical stability of I1 and I2 while increasing that of the variable intermediates and last intermediate. Furthermore, and independent of the buffer condition, there is a gap in the average unfolding forces between I1 and I2 (<25 pN), and the variable intermediates and the last intermediate (>25 pN), which suggests the

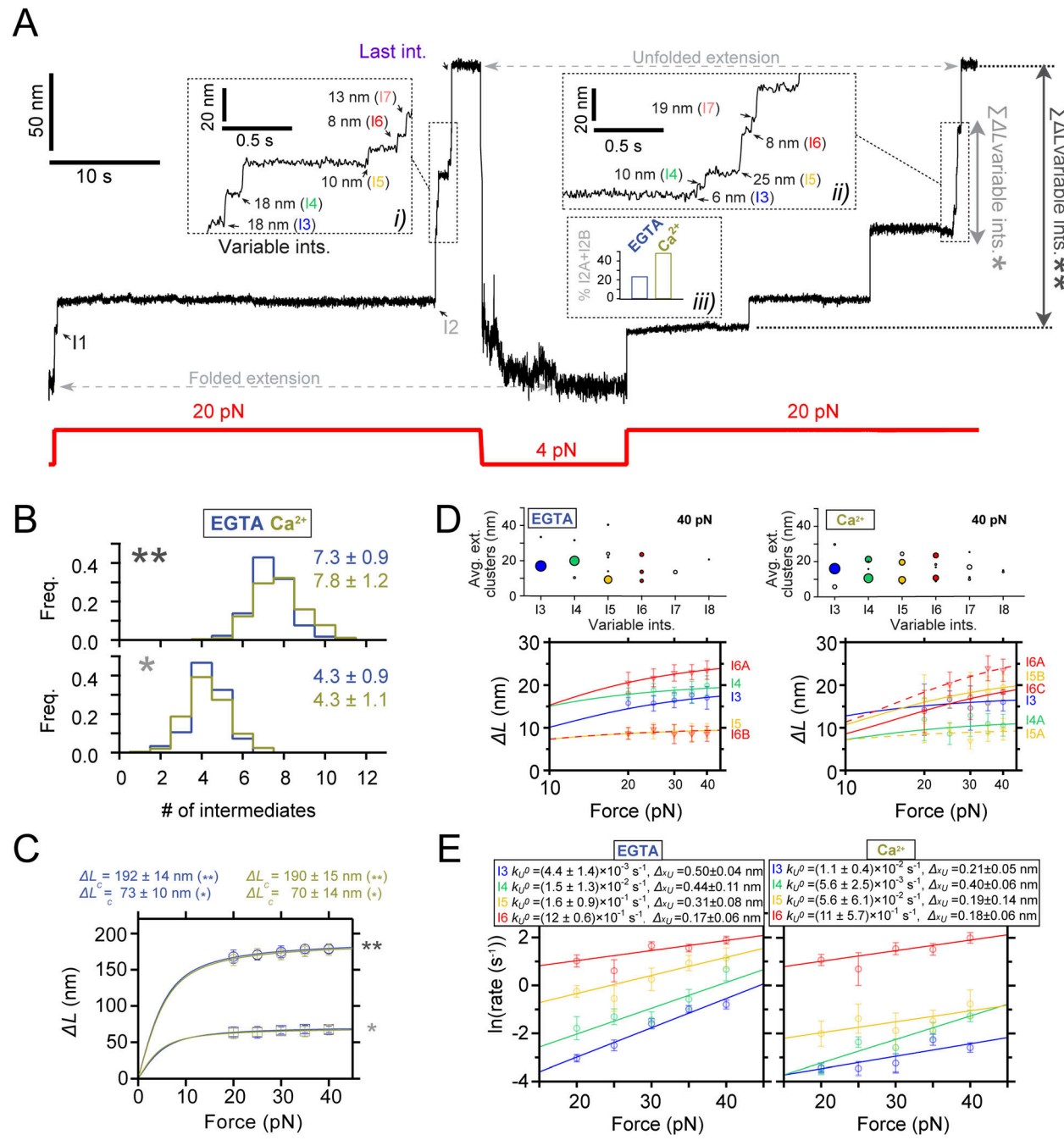

**Figure 2. Ca²⁺-induced conformational heterogeneity and stabilization of PilY1⁵⁰¹⁻¹¹⁶¹.**

**(A)** Magnetic tweezer trajectory of PilY1⁵⁰¹⁻¹¹⁶¹ exposed to unfolding and refolding cycles at different constant forces. From 4 pN, the force is jumped to 20 pN, and the sequential unfolding of PilY1⁵⁰¹⁻¹¹⁶¹ through multiple intermediates is detected (I1→ I2→ variable intermediates → last intermediate). The force is then quenched to 4 pN to allow protein folding, and later, the 20-pN unfolding pulse is repeated, obtaining the same populations of intermediates. Insets *i* and *ii* highlight the conformational heterogeneity of the variable intermediates between consecutive unfolding pulses. In Ca²⁺, the proportion of I2 domains unfolding in two consecutive sub-intermediates doubled compared with EGTA (inset *iii*). Vertical arrows show the total length of the unfolding intermediates (**) and the variable intermediates (*). **(B)** Distribution of the total number of unfolding intermediates in PilY1⁵⁰¹⁻¹¹⁶¹ (top, **) and variable intermediates (bottom, *) observed across trajectories in Ca²⁺ or EGTA. **(C)** Force-dependent extension change of the combined extension of all PilY1⁵⁰¹⁻¹¹⁶¹ intermediates (**) and variable intermediates (*). Lines are fits of the FJC model to the average extension (mean ± SD), and the resulting values of contour length increment ($\Delta L_C$) are shown above the graph for each condition. **(D)** Cluster analysis results of the extension of the variable intermediates per order of unfolding and force in EGTA (left) or Ca²⁺ (right). On the top is plotted the average extension of the events classified inside each cluster identified for each intermediate at 40 pN. The size of the data points is proportional to the number of events in that cluster (most represented ones are colored). Below are shown the force-dependent extension changes (mean ± SD) of the most populated conformations for each intermediate in EGTA (left) and Ca²⁺ (right). Solid and dashed lines are fits of the FJC model to the most populated and second most populated conformations, respectively (see Figs S3C and S4C). The force axis is plotted on a logarithmic scale. **(E)** Unfolding kinetics under force of the variable intermediates in EGTA and Ca²⁺. Each intermediate's unfolding

existence of two mechanostability ranges across the PilY1$^{501-1161}$ structure.

PilY1$^{501-1161}$ contains two $Ca^{2+}$-binding sites, but from these observations, we cannot locate which group of intermediates could have these motifs. We sought to explore the variable intermediate group further, attending exclusively to their unfolding order sequence for their classification of mechanical stabilities. Fig S1C (color coding of the intermediates is unrelated to the colors chosen for Fig 1A and B) and Table S1 show the comparison of the unfolding force means and summarize the results (see the Materials and Methods section and Supplementary Information). In both EGTA and $Ca^{2+}$, the first variable intermediate, I3, shows no differences in its average unfolding force (26.5 ± 0.4 pN to 25.8 ± 0.4 pN). In contrast, $Ca^{2+}$ increases the unfolding forces of the following intermediates, from the second variable intermediate, I4, to the last intermediate. Comparisons inside each buffer condition reveal that in EGTA (Fig S1C, Tables S1 and S2), there is no significant change in the average unfolding forces among the variable intermediates, from I3 to I7 (26.5 ± 0.4 pN to 27.5 ± 1.0 pN), suggesting that these structures have similar stability. With $Ca^{2+}$ (Fig S1C, Tables S1 and S3), there is a significant ~5 pN increase in the average mechanical stability between I3 and I4 (25.8 ± 0.4 pN to 30.4 ± 0.4 pN). After I4 and until the last intermediate, the means show no significant differences, indicating similar stability. These results suggest that the variable intermediate I4 is a plausible candidate to harbor one of the $Ca^{2+}$-binding sites and that the mechanical stabilization shift observed in subsequent intermediates could be due to $Ca^{2+}$ binding on this intermediate. $Ca^{2+}$ specifically induces this effect on the mechanical stability of the protein, though other divalent cations can also alter PilY1's nanomechanics (Fig S2A–C).

## $Ca^{2+}$ binding alters the unfolding pathway

The force-ramp experiments aided us in establishing a benchmark for further testing the nanomechanics of PilY1$^{501-1161}$. We defined a clear mechanical signature of the protein, which consists of a hierarchical unfolding sequence of intermediates whose stabilities are affected by $Ca^{2+}$. We next explored PilY1$^{501-1161}$ unfolding dynamics under constant force. In these measurements, we can obtain a clearer picture of the force-dependent extensions of each of the intermediates, narrowing the variability observed in the force-ramp experiments. Because forces above 20 pN trigger the complete stretching of PilY1$^{501-1161}$, we explored its unfolding dynamics from 20 to 40 pN to capture all the intermediates.

Fig 2A shows a trajectory of PilY1$^{501-1161}$ subjected to unfolding and refolding cycles at constant forces. Changing the force from 4 to 20 pN drives the protein from its folded state to its complete unfolding through multiple intermediates. After, the force is quenched to 4 pN to allow protein folding, which reaches the same folded extension. A subsequent 20-pN pulse reproduces the same pattern of unfolding intermediates. We identified the intermediate groups previously categorized (Figs S3A and S4A). In EGTA or $Ca^{2+}$, the average number of variable intermediates is ~4. The number of total intermediates oscillates around seven to eight, with a slightly higher count when $Ca^{2+}$ is present (Fig 2B). The proportion of I2 unfolding events occurring in two steps doubled in the presence of $Ca^{2+}$ (Fig 2A, inset *iii*), explaining this increase. Fitting the FJC model to the average extensions of I1, I2, and the last intermediate yields similar results to those obtained in the force ramp (Figs S3D and S4D). However, the variable intermediates' heterogeneity persists, as seen in Fig 2A, insets *i* and *ii*. For instance, I3 appears as a ~18-nm extension in the first unfolding pulse, whereas the second shows a ~6-nm step. This change in pattern affects all the variable intermediates except for I6, which shows a similar extension. This variability was detected intra- and intermolecularly and cannot be explained by the reshuffling of the unfolding order of the intermediates between pulses, as their sizes differ. To rule out that the variable intermediates were not properly folding between pulses, we measured in all trajectories the extension of all the unfolding events in PilY1$^{501-1161}$ and the extension of the unfolding events of the variable intermediates. Fig 2C shows FJC fits to the combined extension of all the unfolding events (**), which yields an $\Delta L_c$ ~190 nm with both EGTA and $Ca^{2+}$, and the variable intermediates (*), which collectively produce an $\Delta L_c$ ~70 nm in both conditions. These results indicate that despite the different number of events and extensions, the amount of polypeptide sequence released after unfolding is the same in all trajectories; therefore, these variable intermediates achieve a compact folded state.

Although the variable intermediates' heterogeneity was prominent, a sequence of events repeated more frequently. To explore the existence of a canonical unfolding pathway, we resorted to clustering analysis, which enabled us to classify the variable intermediates based on their extension. This helped us to sort conformations, identify the most common ones, and resolve structural identities (Schönfelder et al, 2016) (top graphs in Figs 2D, S3B, and S4B). Fig 2D bottom plots show FJC fits to the average extension of the most common intermediates in EGTA and $Ca^{2+}$ (Figs S3C and S4C). In EGTA, most events can be assigned to unique conformations and are well described by individual fits. The I6 intermediate deviates from this trend with two alternative conformations, although one predominates (I6A). In contrast, the behavior with $Ca^{2+}$ differs. Beyond the I3 intermediate, the upcoming intermediates show alternative conformations that occur in a similar frequency. For instance, I4 has a predominant configuration (I4A $\Delta L_c$ ~12 nm) that differs from the I4 population seen in EGTA ($\Delta L_c$ ~21 nm). In I5, the most represented conformation also differs (I5B $\Delta L_c$ ~23 nm versus I5A $\Delta L_c$ ~10 nm). For I6, the behavior differs as well in both conditions. These results suggest that $Ca^{2+}$ alters the unfolding pathway of PilY1$^{501-1161}$, changing the conformations of the intermediates after the variable intermediate I3.

We questioned next whether $Ca^{2+}$ alters PilY1$^{501-1161}$ unfolding kinetics. In Fig 2E, we show the force-dependent unfolding kinetics for each intermediate. We used Bell's approximation for bond lifetimes under force to model their unfolding kinetics (Bell, 1978). From fits of the data to this model, we can obtain an extrapolation to zero force of the unfolding rates of each intermediate ($k_U^0$), and

kinetics (mean ± SEM) is fitted using Bell's model for bond lifetimes. Above each plot is shown the value of the parameters determined: unfolding force at zero force ($k_U^0$) and distance to the transition state ($\Delta x_U$).

the distance to the transition state ($\Delta x_U$). The comparison between conditions indicates that $Ca^{2+}$ stabilizes intermediates I4 and I5, slowing their unfolding rates ~2.7 and ~2.9 times, respectively. $Ca^{2+}$ has the opposite effect on I3, accelerating its unfolding ~2.5 times, whereas I6 remains unchanged in both conditions. There is a change in the $\Delta x_U$ of I3 and I5, which decreases in comparison with the EGTA condition. This indicates that $Ca^{2+}$ stabilizes PilY1$^{501-1161}$, which confirms the observations made in force-ramp experiments.

### PilY1$^{501-1161}$ harbors a mechanosensing-like structure

The previous sections described PilY1$^{501-1161}$ nanomechanics at force regimes that promote its complete unfolding. The sequential unfolding of PilY1$^{501-1161}$ suggests that I1 and I2 are less stable than the other intermediates, which divides the protein into two ranges of mechanostability (Fig S5). Therefore, we conducted constant force measurements at loads below 10 pN, to test whether PilY1$^{501-1161}$ exhibits dynamics in this force regime.

Fig 3A shows a trajectory of PilY1$^{501-1161}$ exposed to high and low mechanical loads. After unfolding at high force, the load is quenched to 5.5 pN, which promotes the collapse of the extended polypeptide and enables folding. At this force, PilY1$^{501-1161}$ dwells over time across three levels of extension, which correspond to structure folding and unfolding. The lowest extension level corresponds to the folding of I1, hence the complete folding of PilY1$^{501-1161}$ (Fig 3A, inset *i*). The second and third levels correspond to I1 unfolded extension (and I2 folded extension) and I2 unfolded level, respectively. In Fig 3A, inset *ii*, we show long dynamics of PilY1$^{501-1161}$ at forces ranging from 4.5 to 6.0 pN, from where it can be seen how the increase in the mechanical load tilts the equilibrium toward the extended (unfolded state) levels of I1 and I2. Measuring at this force range allowed us to determine the force-dependent step size of I1 and I2 below 20 pN (Figs S3D and S4D), which enabled us to assign their identity to these dynamics. From these long equilibrium dynamics (Fig 3A, inset *ii*), we determined the folding probability as a function of the force for I1 and I2 with EGTA or $Ca^{2+}$ (Fig 3B). As the mechanical load rises, the folded fraction of both intermediates decreases, adopting a sigmoidal shape. The force value at which the intermediates spend the same amount of time in the folded and unfolded state is defined as $P_{50}$. $Ca^{2+}$ shifts $P_{50}$ to lower values for I1 (4.7 to 4.4 pN) but increases it for I2 (5.7 to 5.9 pN). Moreover, $Ca^{2+}$ changes the force dependency of I2 folding probability, which is broadened and spans 3 pN (from fully folded at 4 pN to unfolded at 7 pN), 0.5 pN more than with EGTA (4.0–6.5 pN).

We also tested whether the kinetics of I1 and I2 were affected by $Ca^{2+}$. Fig 3C shows both intermediates' folding and unfolding kinetics in EGTA or $Ca^{2+}$. Using Bell's model, we observe that the folding and unfolding kinetics of both I1 and I2 reproduce the observations from the folding probability. From the extrapolations to zero force, in $Ca^{2+}$ I1 unfolds and folds faster (~1 time and ~1.9 times) than in EGTA. I2 folding rate at zero force is slowed down in $Ca^{2+}$ by ~100 times, which comes from a steep force-dependent change (EGTA $\Delta x_F = -10.82 \pm 0.18$ nm versus $Ca^{2+}$ $\Delta x_F = -6.85 \pm 0.44$ nm), whereas its unfolding rate is slowed down ~1.4 times in comparison with the EGTA condition.

PilY1$^{501-1161}$ dynamics at low force reveal that the hierarchy is conserved for I1 and I2, and both exhibit a $Ca^{2+}$-tuned mechanical sensitivity in a narrow range of forces, which showcases the prototypical behavior of mechanosensors. These results indicate that at least one of them contains one $Ca^{2+}$-binding site, although both are affected by metal binding to this motif.

### The integrin-binding domain is the mechanosensitive structure of PilY1$^{501-1161}$

After demonstrating the mechanosensing-like behavior of PilY1$^{501-1161}$, we wanted to link the observed intermediates with structural modules of this protein. The PilY1$^{501-1161}$ sequence contains the β-propeller fold and a stretch of 141 residues in the N terminus that contains the RGD and $Ca^{2+}$-binding motifs responsible for integrin binding. Unlike the β-propeller, the structure of this section from PilY1 has not been resolved (Fig 1B shows a prediction from AlphaFold [Jumper et al, 2021; Varadi et al, 2022]).

To elucidate which of the intermediates of PilY1$^{501-1161}$ corresponds to this sequence containing the RGD and first $Ca^{2+}$-binding motifs, we synthesized a second construct that only contains the β-propeller, named PilY1$^{642-1161}$ (Fig 4A). Fig 4B shows a trajectory of PilY1$^{642-1161}$ exposed to unfolding and refolding pulses. The unfolding trajectories show no sign of I1 and I2, but only the variable intermediate and last intermediate groups (Figs 4B, S6A–D, and S7A–D). The average number of total intermediates drops to ~5, whereas the number of variable intermediates remains unchanged (Fig 4C), which indicates the loss of two domains. The complete unfolded extension of all intermediates (Fig 4D) is ~60 nm shorter than PilY1$^{501-1161}$ (Fig 2C), but the variable intermediates' complete extension is similar. After cluster analysis (Figs S6B and S7B), the variable intermediates reproduce the conformational structures observed in PilY1$^{501-1161}$ (Figs 4E and F, S6C, and S7C), and the effect that $Ca^{2+}$ has in both the unfolding pathway and kinetics (Fig 4G, and Table S4). Therefore, the second $Ca^{2+}$-binding site, located in blade IV of the β-propeller, is responsible for mechanically stabilizing the variable intermediates. Together with the results obtained in Fig 2, the intermediate I4 seems the most plausible candidate for being blade IV in the β-propeller.

These findings demonstrate that the sequence spanning residues 501–641 is involved in the dynamics observed at 4.0–6.5 pN. However, the removal of this 140-long residue sequence has a larger-than-expected impact on the total length of the protein. The predicted structure (Fig 4H) indicates that the N and C termini of this integrin-binding domain are close to each other ($L_0$ ~1 nm). Assuming a length of 0.36 nm·residue$^{-1}$, the 140-long polypeptide would yield an $\Delta L_c$ ~ 49 nm, in agreement with the values obtained for I2 (Figs 4H, S3D, and S4D). Although these observations support the connection between the 501–641 sequence and I2, the missing ~60-nm length of PilY1$^{641-1161}$ accounts for the loss of I2 and I1, which was never detected. Measurements of the complete extension of the PilY1$^{501-1161}$ and PilY1$^{642-1161}$ constructs from 0.5 pN to its full unfolding at 40 pN reveal that both proteins achieve their full predicted extensions based on their sequence length (Fig S8A–C). Hence, the absence of the I1 signature in PilY1$^{642-1161}$ could be due to its inability to fold properly if the integrin-binding domain is not present.

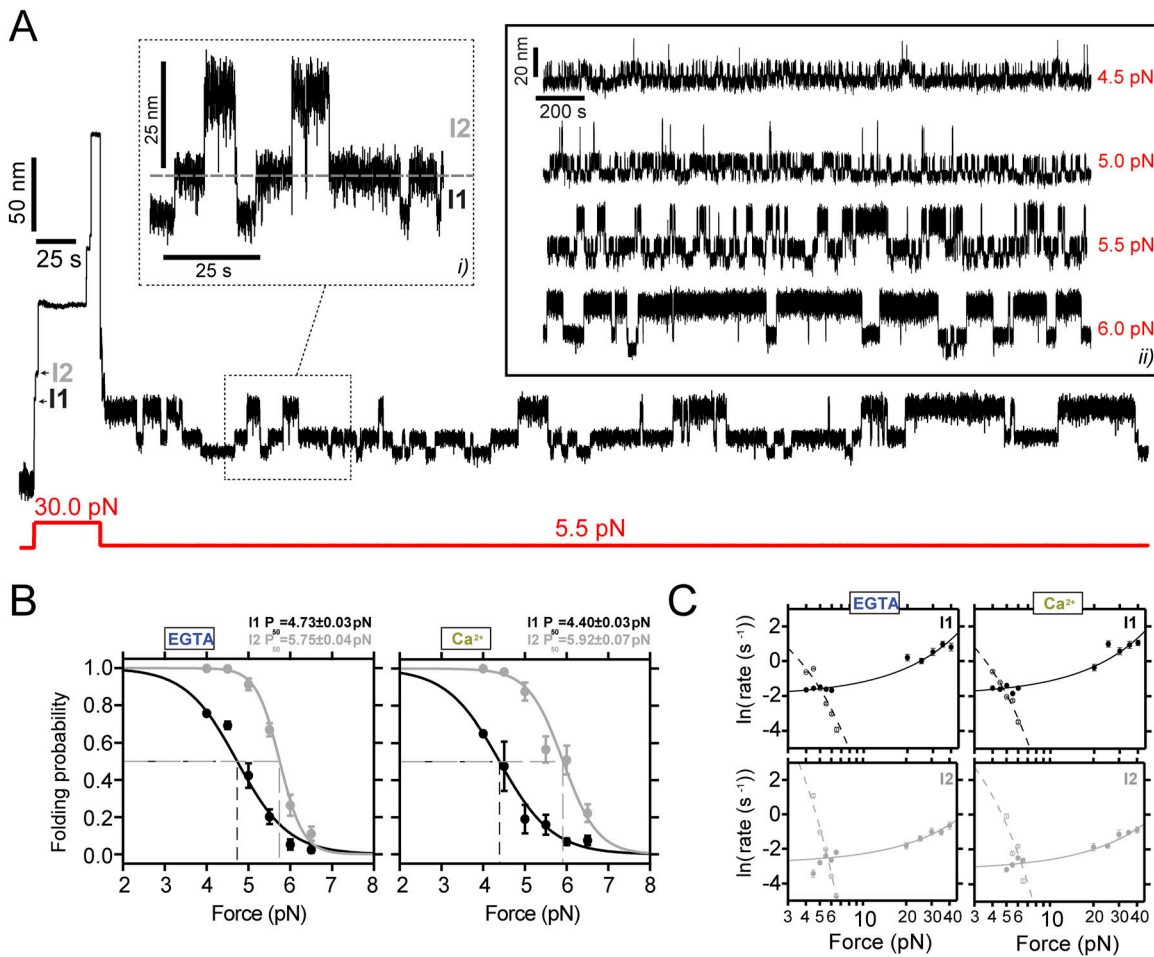

**Figure 3. Conformational dynamics of PilY1$^{501-1161}$ at low force reveal a mechanosensing-like behavior.**
**(A)** Magnetic tweezer trajectory of PilY1$^{501-1161}$ exposed to an unfolding pulse at 30 pN followed by a long 5.5-pN pulse. As shown before, the high-force pulse induces the unfolding of PilY1$^{501-1161}$ through multiple intermediates. At low force, most of the protein refolds; however, part of the protein dwells across three levels of extensions (inset *i*) corresponding to the folded extension of I1 (lowest), the unfolded I1/folded I2 extension (middle), and unfolded extension of I2 (highest). The unfolding hierarchy previously observed at high forces is maintained in this force regime for I1 and I2, which also applies to their folding. Inset *ii* shows ~2,000-s-long trajectories of the dynamics of I1/I2 under different forces. The increase in the mechanical load promotes the occupation of the extended levels. **(B)** Folding probability of I1 and I2 between 4 and 6.5 pN in EGTA and Ca$^{2+}$. The folded fraction of both intermediates sharply decreases with force, following a sigmoidal trend (line fits). The coexistence force ($P_{50}$) of I1 shifts to lower values, whereas that of I2 increases in the presence of Ca$^{2+}$. Ca$^{2+}$ also broadens the force range at which I2 exhibits folding and unfolding dynamics (~2.5 pN in EGTA and ~3 pN in Ca$^{2+}$). **(C)** Folding and unfolding kinetics under force of I1 and I2 in EGTA and Ca$^{2+}$. The logarithm of the rate (mean ± SEM) is plotted against the force in logarithmic scaling. I1 and I2 unfolding and folding kinetics are described with Bell's model (fit lines) in both conditions (I1, empty circles and dashed lines for folding: EGTA $k_F^0$ = [7.1 ± 0.7] ×10 s$^{-1}$, $\Delta x_F$ = − 4.83 ± 0.09 nm; Ca$^{2+}$ $k_F^0$ = [13.4 ± 2.5] ×10 s$^{-1}$, $\Delta x_F$ = − 5.62 ± 0.16 nm. Solid circles and lines for unfolding: EGTA $k_U^0$ = [13.7 ± 0.4] ×10$^{-2}$ s$^{-1}$, $\Delta x_U$ = 0.33 ± 0.01 nm; Ca$^{2+}$ $k_U^0$ = [14.4 ± 0.4] ×10$^{-2}$ s$^{-1}$, $\Delta x_U$ = 0.34 ± 0.01 nm; I2 empty circles and dashed line for folding: EGTA $k_F^0$ = [2.4 ± 0.6] ×10$^5$ s$^{-1}$, $\Delta x_F$ = − 10.82 ± 0.18 nm; Ca$^{2+}$ $k_F^0$ = [2.3 ± 1.4] ×10$^3$ s$^{-1}$, $\Delta x_F$ = − 6.85 ± 0.44 nm. Solid circles and lines for unfolding: EGTA $k_U^0$ = [5.8 ± 0.2] ×10$^{-2}$ s$^{-1}$, $\Delta x_U$ = 0.23 ± 0.01 nm, Ca$^{2+}$ $k_U^0$ = [4.2 ± 0.2] ×10$^{-2}$ s$^{-1}$, $\Delta x_U$ = 0.24 ± 0.01 nm).

## Integrin binding blocks the unfolding of I2

The evidence from the previous section suggests that the intermediate I2 could be the putative structure that contains the RGD and Ca$^{2+}$-binding site responsible for integrin binding. To test this hypothesis, we conducted binding experiments in the presence of the extracellular domains of human integrin αVβ5 (Fig 5A), which have shown the integrin with the highest affinity to PilY1 (Johnson et al, 2011). To conduct these experiments, we used a buffer supplemented with calcium, magnesium, and manganese (binding buffer), because these metals are important for the structural integrity of integrin (Smith et al, 1990; Johnson et al, 2011). Before proceeding with the

binding experiments, we characterized PilY1$^{501-1161}$ nanomechanics in this new buffer. Figs S9A–C and S10A–G summarize these results in force-ramp and constant force measurements, respectively. The protein exhibits a mixed mechanical behavior, where the unfolding force means of I1 and I2 are like the ones observed with EGTA, but the variable and last intermediates reproduce the Ca$^{2+}$ condition results (Fig S9A–C). This can be explained due to the presence of these three divalent cations and their influence on PilY1 nanomechanics (Fig S2A–C), where Mn$^{2+}$ and/or Mg$^{2+}$ could be interfering with Ca$^{2+}$ binding in the integrin-binding domain. In contrast, the unfolding force means of the intermediates modulated by the second Ca$^{2+}$-binding site are the same as those under the Ca$^{2+}$ buffer condition,

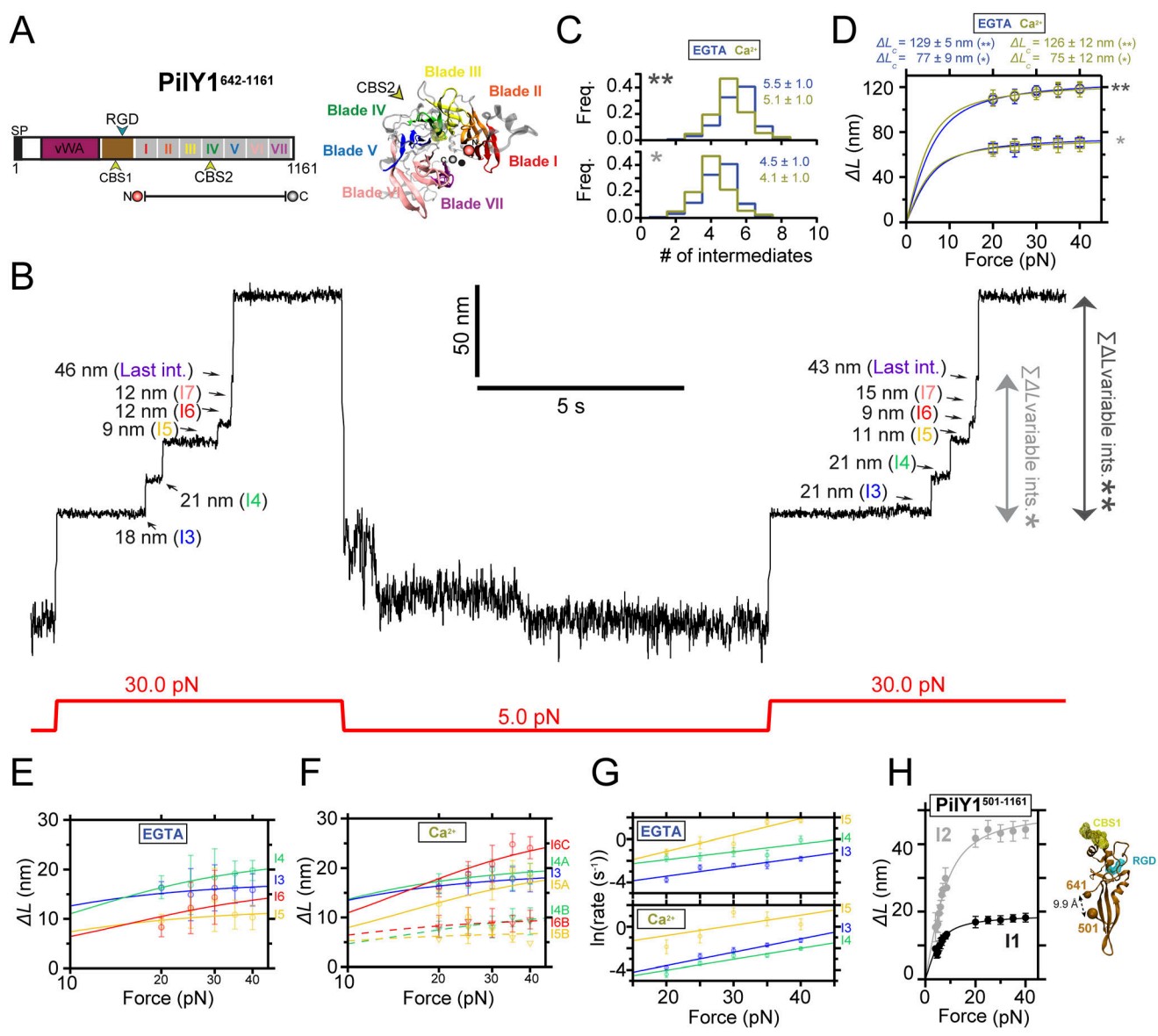

**Figure 4. Nanomechanics of PilY1$^{642-1161}$ indicates that the integrin-binding domain is responsible for the mechanosensor-like behavior observed at low force.**
**(A)** Scheme of PilY1 protein and crystal structure (PDB:3HX6 [Orans et al, 2010]) of the sequence employed spanning residues 642–1,161. **(B)** Magnetic tweezer trajectory of PilY1$^{642-1161}$ exposed to unfolding and folding cycles. Removing the sequence 501–641 yields a mechanical fingerprint identical to PilY1$^{501-1161}$ but lacking the intermediates I1 and I2, where we only detect the variable intermediate and last intermediate groups. **(C)** Distribution of the total number of unfolding intermediates in PilY1$^{642-1161}$ (top, **) and variable intermediates (bottom, *) observed across trajectories in Ca$^{2+}$ or EGTA. **(D)** Force-dependent change in the combined extension of all PilY1$^{642-1161}$ intermediates (**) and variable intermediates (*). Lines are fits of the FJC model to the average extension (mean ± SD), and the resulting values of contour length increment (Δ$L_c$) are shown above the graph for each condition. **(E, F)** Force-dependent extensions (mean ± SD) of the most populated conformations determined with cluster analysis for each intermediate in EGTA (E) and Ca$^{2+}$ (F). Solid and dashed lines are fits of the FJC model to the most populated and second most populated conformations, respectively (see Figs S6C and S7C). The force axis is plotted on a logarithmic scale. **(G)** Unfolding kinetics under force of the variable intermediates in EGTA and Ca$^{2+}$. Each intermediate's unfolding kinetics (mean ± SEM) is fitted using Bell's model for bond lifetimes (see Table S4). **(H)** Based on its sequence length and predicted structure, the integrin-binding domain could be the I2 intermediate observed in PilY1$^{501-1161}$, which yields an Δ$L_c$ ~50 nm (plot on the left). In contrast, I1 would not be a good candidate to hold the integrin-binding site because of its smaller extension.

suggesting that both sites could have different affinities for various divalent cations.

After setting the basis of PilY1$^{501-1161}$ nanomechanics in binding buffer, we conducted integrin-binding experiments under force. Fig 5B details the fingerprint for integrin-binding events. At 6 pN, I1 and I2 domains fold and unfold in equilibrium, visiting the three different levels of extension. As shown in the inset, the protein stops visiting the

unfolded extension of I2 at the moment indicated with a green arrow, but the dynamics continue unchanged below this level. This shortening of the I2 unfolded extension only occurs and is specifically induced in the presence of integrin (Fig S11A and B). In Fig 5C, we show a trajectory of PilY1$^{501-1161}$ held at 5 pN for an extended period of time in the presence of integrin. Insets *i* and *iv* indicate binding events and inset *iii* one unbinding event. In the bound state, the I2 domain cannot

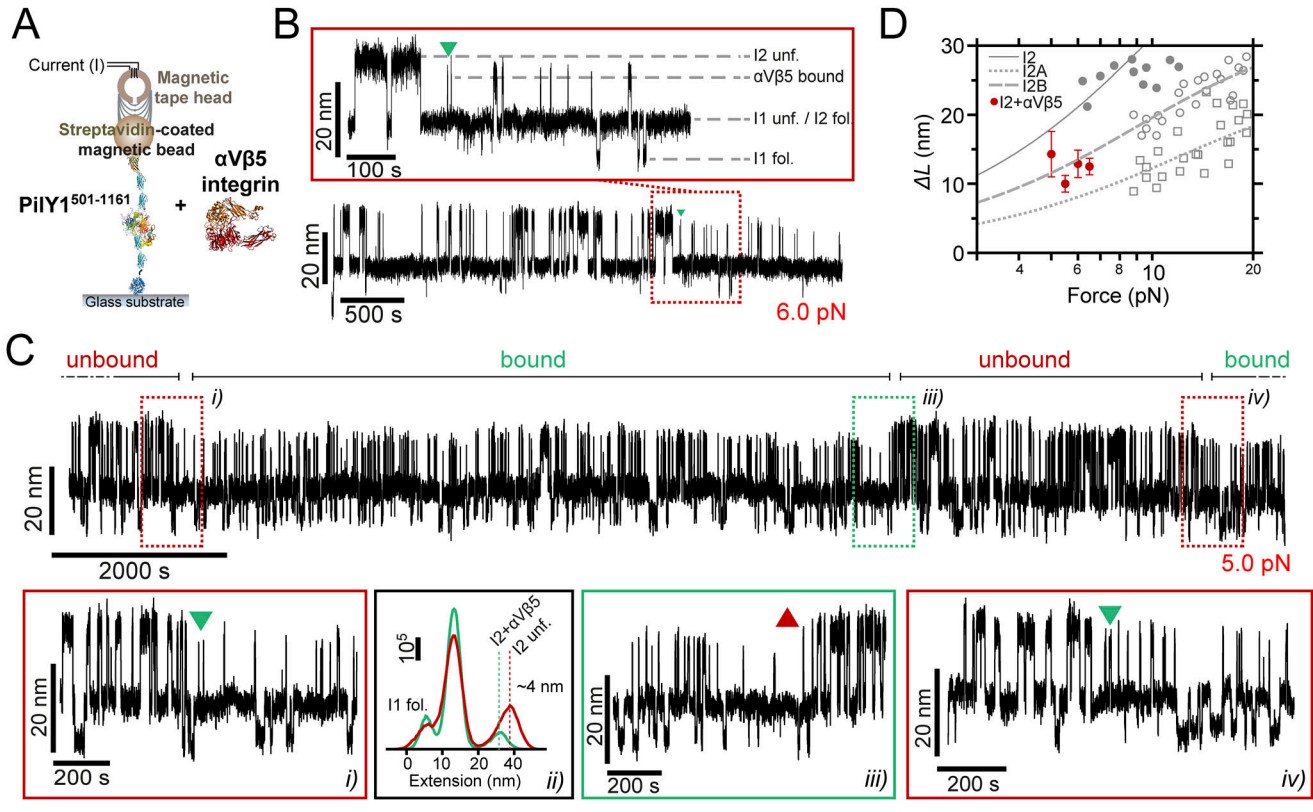

**Figure 5. Integrin binds to I2 in PilY1$^{501-1161}$ under force and prevents its full unfolding.**
**(A)** Diagram of the magnetic tweezer-binding experiment with PilY1$^{501-1161}$ and the extracellular domains of the human integrin αVβ5 heterodimer. **(B)** Magnetic tweezer trajectory of PilY1$^{501-1161}$ held at 6 pN in the presence of integrin. I1 and I2 domains fold and unfold in equilibrium, and integrin binding is detected as a shortening of the unfolded extension of I2 (inset). **(C)** ~4-h-long trajectory at 5 pN where integrin binds (insets *i* and *iv*) and unbinds (inset *iii*) to PilY1$^{501-1161}$ I2 domain, yielding a ~4-nm change in the extension. Inset *ii* shows the extension distribution of the protein before binding (red) and after binding (green). The bar shows the number of points. **(D)** Comparison of the force-dependent extension changes of the integrin-bound I2 (mean ± SD) with force-ramp data from non-bound I2 and its two sub-intermediates I2A and I2B. Fitting the data with the FJC model reveals that integrin-bound I2 force-dependent extension falls within the fit for I2B sub-intermediate, which indicates that integrin binding prevents the unfolding of I2A.

reach its unfolded extension, but a shorter one. After unbinding, I2 recovers its corresponding extension at the measured force (inset *ii* shows the protein extension distribution before and after binding). We measured the average extension of this new conformation of I2 across the forces where integrin binding was detected (Fig 5D). When compared to the force-dependent extension of naive I2, we can see that the integrin-bound I2 average extension falls within the FJC fit of the sub-intermediate I2B. This would indicate that integrin binding partially blocks the unfolding of I2 and that the sub-intermediate I2A cannot extend under this condition. Integrin recruitment occurs only in the presence of the RGD motif (residues 617–619) present in the PilY1$^{501-1161}$ construct (Johnson et al, 2011), which bestows the specificity of the PilY1–integrin interaction (Fig S12). These observations indicate that I2 is the conformational structure containing the integrin-binding domain, which spans residues 501–641.

# Discussion

Bacteria colonize and adhere to substrates upon recognition of the mechanical stimuli elicited by surface proximity (Rodesney et al,

2017; Wang et al, 2023). These cues trigger signaling pathways in free-swimming cells that promote their attachment to target substrates and the formation of biofilms. This change of lifestyle reprograms gene expression and, in pathogenic bacteria, kick-starts the deployment of virulence factors, having important implications for infection progression (Heiniger et al, 2010; Luo et al, 2015).

The ability of bacteria to detect mechanical forces has been linked to their motor-driven appendages, the flagella, and the T4P (O'Toole & Kolter, 1998; Lele et al, 2013). The T4P mediate twitching motility and are essential for the surface-attached life of bacteria. PilY1 plays a crucial part in T4P biogenesis and functions, while also serving as a mechanosensor and adhesive protein. Because of its location at the tip of the T4P, PilY1 carries out its functions while exposed to mechanical stress. Here, we have explored how ligand binding and mechanical forces impact PilY1 dynamics.

The sequence comprising the residues 501–1,161 of PilY1 contains ligand-binding motifs that control T4P dynamics and host adhesion. Most of this sequence adopts a 7-bladed β-propeller fold, where blade IV contains the Ca$^{2+}$-binding site that regulates T4P extension and retraction cycles. Forces above 20 pN lead to the sequential hierarchical unfolding of the domains of this

heteropolyprotein. Under force, a polyprotein comprised of independent domains of varying levels of stability will usually undergo the unfolding of the least stable domains first before the more stable ones (Li et al, 2000). In the PilY1 constructs studied (Figs 1, 2, 3, 4, 5, S1, S2, S3, S4, S5, S6, S7, S8, S9, S10, S11, and S12), we observe a tightly conserved unzipping pattern where the unfolding of each domain depends on the unfolding of the previous one in the sequence, which reminds a Matryoshka doll–like configuration. This suggests the existence of long-range contacts and allosteric interactions over the entire structure of the protein (Batey & Clarke, 2006; Perez-Riba et al, 2018; Santorelli et al, 2023). The apparently conserved unfolding pathway is challenged by the heterogeneous conformations exhibited by the variable intermediate group. $Ca^{2+}$ binding to its cognate site in blade IV increases the mechanical stability of the variable intermediate I4 and delays the unfolding of the subsequent domains (Figs 2 and 4, S1, and S9). Furthermore, this metal increases the frequency of conformations sparsely visited in EGTA (Figs 2 and 4, S3, S4, S6, and S7), indicating that metal binding impacts the mechanical properties and the unfolding pathway of the protein (Cao et al, 2011; Scholl et al, 2016; Löf et al, 2019; Yuan et al, 2019; Nie et al, 2022). The fact that the total unfolding extension of the variable intermediates is always the same confirms that this region of PilY1 folds between unfolding cycles (Figs 2 and 4). A source of conformational variability could be that after quenching the force and allowing folding, the variable intermediates experience misfolding events where their contacts are not achieved intradomain, but instead, non-native interactions are established with neighboring modules (Oberhauser et al, 1999). If that was the case, the unfolding rates of the variable intermediates could not be explained as single processes (Figs 2 and 4), but as a multitude of kinetic processes, each departing from a different folded state. Hence, the hierarchical pattern observed for I1, I2, and the last intermediate groups also applies inside the variable intermediate sequence.

Our findings indicate that the PilY1 $\beta$-propeller has a partially conserved unfolding pathway that $Ca^{2+}$ alters when bound to blade IV. A rough estimation of the expected unfolded extension of each blade of the $\beta$-propeller and its candidate intermediates is shown in Fig S13. PilY1 is not a canonical $\beta$-propeller fold (Mylemans et al, 2021), and blades V-VII have a different number of strands than blades I-IV, which are four-stranded antiparallel $\beta$-sheets. However, the estimated contour length of all blades is similar, and assigning their identities is not straightforward. Furthermore, blades V and VI share one strand (Orans et al, 2010), which makes it even more challenging to devise what kind of unfolding intermediate or intermediates could generate. Ultimately, the core population of the variable intermediates could be narrowed down to four different structures of $\Delta L_c$ ~10–20 nm (I3 to I6), and the alternative conformations observed can be explained as the rupture of substructures spanning different blades.

From a biological standpoint, the stabilizing effect of $Ca^{2+}$ in the $\beta$-propeller could be relevant in T4P dynamics regulation. *P. aeruginosa* T4P exerts pulling forces ~30 pN that overlap with the force ranges we have tested and where $Ca^{2+}$ has shown its stabilizing effect (Ribbe et al, 2017). How the $Ca^{2+}$-bound state of PilY1 antagonizes T4P retraction is not yet clear, but one study hypothesized that PilY1, in association with other pilus-tip proteins involved in T4P biogenesis, might block the full retraction of the T4P beyond the outer membrane PilQ secretin pore, acting as a cork. In this scenario, $Ca^{2+}$-induced mechanical stabilization of PilY1 would maintain the bulky structure of the $\beta$-propeller and prevent its unfolding, blocking further pilus retraction. This would ensure the structural integrity of the pilus-tip complex for priming pilus polymerization in a subsequent extension cycle (Treuner-Lange et al, 2020; Guo et al, 2024). It is also feasible that the increased mechanical stabilization of PilY1 by $Ca^{2+}$ could be linked to its adhesive role, contributing to the structural integrity of this protein during surface attachment, as has been proposed for other bacterial pilin proteins (Echelman et al, 2016; Oude Vrielink et al, 2017; Milles et al, 2018). Across species, it has been seen that PilY1 orthologs exhibit different $Ca^{2+}$ affinities, and its effect in T4P dynamics varies from *P. aeruginosa*. This would entail that the $Ca^{2+}$-mediated binding and T4P regulation could be finely tuned to the chemical environment of bacteria (Morand et al, 2004; Cruz et al, 2014; Parker et al, 2015). Similar to chemical affinity, the nanomechanics of PilY1 is probably adapted to the environment and are specific to the host. This specificity would create a unique interplay between mechanical forces and ligand binding in regulating PilY1 functions, depending on the substrate being colonized (Cruz et al, 2012, 2014; Parker et al, 2015; Herfurth et al, 2022; Hernández-Sánchez et al, 2024).

PilY1 mechanical behavior at high force can match its expected function as a T4P retraction inhibitor, but its dynamics in the low-force regime resemble a force-sensing activity linked to its adhesin function. The I2 intermediate seems the most plausible candidate to harbor the motifs responsible for integrin binding, not only by its dimensions but also for the signature conformation it adopts upon binding (Figs 3, 4, and 5). Shortening of the unfolded extension of proteins after ligand binding has been reported in the proteins talin and vinculin, keystones of eukaryotic mechanotransduction (Yao et al, 2014; Tapia-Rojo et al, 2020). Integrin binding to PilY1 prevents the full extension of the I2 domain, but, unlike the talin–vinculin interaction, the remaining structure of I2 can still undergo folding and unfolding transitions. The conformation of this halfway structure matches the force-dependent extension of the sub-intermediate I2B; hence, integrin binding prevents the unfolding of I2A. These sub-intermediates unfold almost simultaneously in a sequence I2A-I2B and are noticeable when pulling at high forces. Their frequency increases with $Ca^{2+}$ (Figs 2 and S10), which suggests that metal binding in I2 could promote a structural organization of the binding pocket that facilitates integrin recruitment (Johnson et al, 2011). Furthermore, $Ca^{2+}$ binding in I2 seems to activate an allosteric mechanism that lowers the stability of I1, which might have implications for integrin-binding modulation under tensile stress. This structural connection between I1 and I2 seems plausible because the elimination of I2 abrogates the mechanical signature of I1 (Fig 4), pointing out that both domains might be interdependent for folding. This force and posttranslational modulation of the nanomechanics of the integrin-binding domain and I1 could entail an evolutionary tuning to the mechanical environment of the respiratory epithelium, as it has been proposed for other bacterial adhesive structures (Alonso-Caballero et al, 2018, 2021).

Integrin binding has been shown to occur in the absence of force (Johnson et al, 2011); nevertheless, the allosteric interactions that seem to govern PilY1 mechanics could play a significant role in vivo, promoting or hindering host adhesion depending on the mechanical environment experienced by the bacterium.

Overall, our approach to PilY1 dynamics provides mechanistic details of how force could tune some of its functions at the nanoscale. We propose that PilY1 functions are mechanically compartmentalized; their recruitment depends on the stability of the structures that harbor them, ligand binding, and the mechanical load. Future work should address the nanomechanics of the vWA domain, given its central role in bacterial mechano-transduction and its spatial location in PilY1. Structural models indicate that the vWA domain may be blocking the integrin-binding site, meaning this site might only become accessible for ligand binding after the domain undergoes force-induced conformational changes (Guo et al, 2024). Our findings with the C-terminal section of PilY1 suggest a complex network of interactions among structural modules, and the multiple functions of PilY1 under force may depend on its overall structure.

# Materials and Methods

## Cloning and expression of chimeric constructs

All reagents employed were purchased from Sigma-Aldrich, unless otherwise indicated. The sequence of the proteins PilY1$^{501–1161}$ and PilY1$^{642–1161}$ from the *P. aeruginosa* PAO1 strain was optimized for expression in *Escherichia coli*. Both sequences were ordered (GeneArt; Thermo Fisher Scientific) with a 5′-end Kpn2I and a 3′-end NheI restriction sites, which enabled their cloning into modified pFN18A (HaloTag) T7 Flexi Vector (Promega). The expression cassette contained, from 5′- to 3′-end, an N-terminal HaloTag protein, four copies of the human titin I32 domain, HisTag sequence, and AviTag sequence. The sequence between the second and third I32 domain gene copies contained the Kpn2I and NheI restriction sites, which allowed the insertion of the proteins of interest in sense. Cloning and DNA amplification procedures were carried out in *E. coli* TOP10 strain (Thermo Fisher Scientific), and sequence fidelity was confirmed (GeneWiz).

Protein expression (PilY1$^{501–1161}$ construct) was conducted in *E. coli* BL21 Star strain (Thermo Fisher Scientific). Cell culture was grown to OD$_{600}$~0.6 in LB broth containing 100 $\mu$g·ml$^{-1}$ carbenicillin at 37°C and 250-rpm shaking, and then, 1 mM IPTG (Invitrogen) was added to induce protein expression at 25°C for 3 h and at constant shaking. After, cells were pelleted (4°C, 4,000$g$, 20 min) and resuspended in wash buffer (50 mM NaPi at pH 7.0, 300 mM NaCl, 20 mM imidazole, 10% vol/vol glycerol) supplemented with EDTA-free protease inhibitor (Roche). Cells were then incubated for 30 min on ice with 1 mg·ml$^{-1}$ lysozyme, 5 $\mu$g·ml$^{-1}$ DNase I, 5 $\mu$g·ml$^{-1}$ RNase A, and 10 mM MgCl$_2$. Then, the cells were mechanically lysed in a French press (G. Heinemann) and cell debris was pelleted (4°C, 40,000$g$, 1 h). The supernatant was incubated with HisPur Cobalt resin (Thermo Fisher Scientific) at 4°C for 1 h on an orbital shaker. The His-tagged proteins were eluted from the resin with wash buffer containing 250 mM imidazole and subsequently biotinylated in vitro at room temperature for 1 h. After, the protein solution was further purified by size-exclusion chromatography in a Superdex 200 Increase 10/300 GL column (Cytiva). The protein was eluted from the column in storage buffer (10 mM Hepes, pH 7.2, NaCl 150 mM, 1 mM EDTA, 10% vol/vol glycerol), aliquoted, snap-frozen in liquid N$_2$, and stored at −80°C until use. The PilY1$^{642–1161}$ construct was expressed in *E. coli* T7 Express strain (New England Biolabs) and cotransformed with the pBirAcm plasmid (Avidity) to biotinylate the protein in vivo. Cell culture was grown to OD$_{600}$~0.6 in LB broth containing 100 $\mu$g·ml$^{-1}$ carbenicillin and 10 $\mu$g·ml$^{-1}$ chloramphenicol (pBirAcm). Protein expression and biotinylation were induced with 1 mM IPTG and 50 $\mu$M D-biotin (Invitrogen) for 3 h at 25°C and constant shaking. Downstream purification steps were carried out as described above. Protein purification was assessed using SDS–PAGE (Fig S14).

## Single-molecule magnetic tweezer measurements

Single-molecule experiments were conducted on custom-built magnetic tweezer microscopes, using either a configuration consisting of a pair of permanent magnets, as previously described (Popa et al, 2016), or a magnetic tape head (Tapia-Rojo et al, 2024). In the magnetic tape head configuration, the force is applied to the magnetic bead–bound proteins with a magnetic tape head (Brush Industries), which is fixed on a custom-designed CNC aluminum piece (named card reader). In the base of the card reader lies the experimental fluid chamber where the protein under study is immobilized on the bottom glass (see below for details). The 150-$\mu$m thickness of the bottom glass places the sample-located surface at a 300-$\mu$m distance from the 25-$\mu$m gap of the magnetic tape head, which establishes a known current–force relationship (Tapia-Rojo et al, 2019). The 25-$\mu$m gap of the magnetic tape head is aligned with the optical path of the microscope. The base of the card-reader piece contains an opening to accommodate a 100X oil-immersion Plan Apochromat objective (Zeiss) that enables the visualization of the reference and magnetic beads located on the bottom glass of the fluid chamber. The objective position was controlled with a P-725 nanofocusing piezo actuator (Physik Instrumente), and the image was acquired with a CMOS camera (Ximea). Image processing was conducted with a custom-written program (C++/Qt) that allowed a ~1.5-kHz image sampling frequency. This software communicated with a NI USB-6289 multifunction DAQ card (National Instruments), which enabled data acquisition and controlling the piezo position and the current supplied (hence, the force applied to the protein) to the magnetic tape head. The magnetic tape head electrical supply came from two 12-V lead–acid batteries, and the current was kept under feedback with a custom-built PID controller. Applying currents in the 0–1,000 mA range allowed the application of forces spanning from 0 to 42 pN using Dynabeads M-270 streptavidin–coated superparamagnetic beads (Invitrogen).

Fluid chamber preparation involved the cleaning and functionalization of a 40 × 24 mm bottom glass (150-$\mu$m thickness) and a 22 × 22 mm top glass (Ted Pella), as previously described

(Popa et al, 2016). After cleaning, the top glass was treated with Repel-Silane for 30 min and the bottom glass was silanized with a 1% vol/vol solution of 3-(aminopropyl)trimethoxysilane in ethanol for 10 min. After functionalization, both types of glasses were rinsed in 100% ethanol, dried with $N_2$, and curated in an oven at 100°C for at least 1 h.

The fluid chambers were assembled by melting a bowtie-shaped parafilm spacer placed between the top glass and the bottom glass at 80°C for 5 min. A laser cutter was used to generate the parafilm spacer shape. After sandwiching between glasses, the parafilm shape created two wells flanking the top glass that allowed buffer exchange during experiments. Further fluid chamber functionalization steps carried out at room temperature involved the sequential treatment with 1% vol/vol solution of glutaraldehyde in 0.1 M phosphate buffer (PBS) for 1 h, covalent immobilization of a 0.05% wt/vol solution of amino-functionalized polystyrene reference beads (2.5–2.9 μm in diameter; SpheroTec) in PBS for 20 min, extensive rinse with PBS, and covalent anchoring of 20 μg·ml$^{-1}$ HaloTag amine ligand (Promega) in PBS overnight. After HaloTag ligand functionalization, the fluid chambers were passivated for at least 3 h with blocking buffer containing 1% wt/vol sulfhydryl-blocked BSA (Lee Biosolutions) in 20 mM Tris–HCl, pH 7.4, 150 mM NaCl, and 0.01% wt/vol $NaN_3$.

To conduct force experiments, the fluid chambers were incubated with a ~3 nM solution of the chimeric constructs (PilY1[501–1161] and PilY1[642–1161]) at room temperature for 30 min. The unbound protein was removed by flowing buffer from one well of the fluid chamber and removing from the other. The basic composition of the buffer used in the experiments contained 10 mM Hepes, pH 7.2, 150 mM NaCl, and 10 mM ascorbic acid. Depending on the condition, this buffer contained either 2 mM EGTA, 2 mM $CaCl_2$, or binding buffer (2 mM $CaCl_2$, 1 mM $MgCl_2$, and 1 mM $MnCl_2$). The fluid chamber was then fixed with double-sided tape (Tesa) to a custom-made metal stamped fork. The unit formed by the fork-fluid chamber assembly was placed in the base of the card-reader piece, leaving the fluid chamber below the magnetic tape head. The fork was then attached to an XY linear stage (Newport), which allowed to move the fluid chamber and scan different areas during the experiments.

The experiments started by flowing a 1:30 dilution of streptavidin-coated superparamagnetic beads, previously passivated with blocking buffer at 4°C, for >1 h, and constant rotation at 5 rpm. The beads and the surface-immobilized proteins were allowed to react for 2 min in the absence of force. Then, a constant force of 2–4 pN was applied to remove unspecific-bound beads, and the fluid chamber was further flowed with experimental buffer. From here, the different force protocols described in the work were applied. At the forces tested, the unfolding probability of the HaloTag or I32 domains remains low. In the rare event of unfolding, their identity can be easily ascertained and does not alter PilY1 behavior (Fig S15). For the integrin-binding experiments, PilY1[501–1161] proteins were held at forces that allowed folding and unfolding transitions of the I1 and I2 intermediates, and then, a binding buffer solution containing 200 nM of recombinant αVβ5 human integrin extracellular domains (R&D Systems) was slowly added to one of the wells of the fluid chamber, while simultaneously removing the previous buffer volume from the other well until complete buffer substitution.

## Analysis

All the analysis was carried out with Igor Pro 8.0 software (WaveMetrics). Magnetic tweezer recordings were captured at ~1.0–1.5 kHz frame rate, and trajectories were smoothed with a 151 box-sized fourth-order Savitzky–Golay filter. To assess the significance of the difference between the unfolding forces between buffer conditions (EGTA or $Ca^{2+}$) and between intermediates (Tables S1, S2, and S3), the difference of the means and its uncertainty are computed (from the SEM of the values at each condition). The difference is stated significant when the probability of being zero is smaller than 5% ($|t|>t_{critical}$, being $t_{critical}=2$ for $P < 0.05$) (Trosset, 2009).

$$t = \frac{\mu_1 - \mu_2}{\sqrt{SEM_1^2 + SEM_1^2}}.$$

Step size determination of the unfolding intermediates was determined by measuring the extension before and after the transition, and for the equilibrium dynamics at low force, a multi-Gaussian fit was employed to find the center of the extension levels and measure their distance. The force-dependent extension of the intermediates was fitted to the FJC model for polymer elasticity (Flory, 1956):

$$\Delta L(F) = \Delta L_C \left[ \coth\left(\frac{Fl_K}{k_B T}\right) - \frac{k_B T}{Fl_K} \right],$$

where $\Delta L_C$ is the increment of contour length in nm, $l_K$ is the Kuhn length in nm, and $k_B T$ is the thermal energy as 4.11 pN·nm. Behavior at high forces is dominated by the contour length, whereas the Kuhn length is determined by low-force behavior. Hence, the Kuhn length is only reported for those measurements and intermediates, which exhibit folding and unfolding events in the low-force regime (I1 and I2).

The force-dependent folding probability of the intermediates I1 and I2 was fitted with a sigmoidal function:

$$FP(F) = \left( exp\left(\frac{F - P_{50}}{r}\right) + 1 \right)^{-1},$$

where $P_{50}$ is the coexistence force (in pN) between the folded and unfolded state, and $r$ is the rate in pN$^{-1}$.

Dwell time analysis in the unfolding trajectories was measured from the moment the desired force is achieved (~40 μs) until each of the intermediates unfolds. Because their unfolding is sequential and requires that the previous structure unfolds first, their unfolding times were corrected based on this. The same procedure was used for the dynamics registered at low force. Because of their long nature, a custom-written procedure was written to automatically detect the extension levels and assign the folding and unfolding dwell times of I1 and I2 intermediates. Folding and unfolding rates were calculated as the inverse of the average dwell

times, and their force dependency was fitted with the Bell model (Bell, 1978):

$$k_{U/F}(F) = k_{U/F}^0 \, e^{\left( \frac{\pm F \Delta x_{U/F}}{k_B T} \right)},$$

where $k_{U/F}^0$ is the unfolding/folding rate extrapolation to zero force, and $\Delta x_{U/F}$ is the distance to the transition state for unfolding and folding.

Cluster analysis of the variable intermediates was done with R Statistical Software (v 4.3.0; R Core Team, 2023) using the libraries "factoextra" (Kassambara & Mundt, 2020) and "cluster" (Maechler et al, 2024). The k-means algorithm and the Euclidean distance were used to establish the differences in the extension of the conformations of the variable intermediates. We tested several k-means solutions varying the number of clusters from 1 to 20 for each force, protein construct, and buffer condition. We conducted the average silhouette analysis method to find the optimal number of clusters per condition, and this value was used to cluster the data.

## Data Availability

The data supporting the findings of this study are available from the corresponding author upon reasonable request.

## Supplementary Information

## Acknowledgements

The authors thank Pablo Mateos-Gil, Marta Martín, and Rafael Rivilla for helpful discussions and comments. This project has received funding from the European Union's Horizon 2020 research and innovation program under the Marie Sklodowska-Curie grant agreement No. 101028879. This work was supported by and performed at the Francis Crick Institute, which receives its core funding from Cancer Research UK (CC0102), the UK Medical Research Council (CC0102), and the Wellcome Trust (CC0102). FJ Cao-Garcia was supported by the Spanish Ministry of Economy and Competitiveness and the European Regional Development Fund (RTI2018-095802-B-I00), the Spanish Ministry of Science, Innovation and Universities (PID2023-148319NB-I00), and the Universidad Complutense de Madrid (UCM) (PR12/24-31558). JE Walker and S Board were supported by the Leverhulme Trust (RL 2016-015). A Alonso-Caballero and this work were supported by a European Commission Marie Sklodowska-Curie fellowship (NIOBMT-101028879), the project PID2023-146082NA-I00 funded by MICIU/AEI/10.13039/501100011033 and by ERDF/EU, and the grant RYC2021-031965-I (MICIU/AEI/10.13039/501100011033 and European Union NextGenerationEU/PRTR).

### Author Contributions

FJ Cao-Garcia: data curation and formal analysis.
JE Walker: methodology.
S Board: methodology.
A Alonso-Caballero: conceptualization, data curation, formal analysis, validation, investigation, visualization, and methodology.

### Conflict of Interest Statement

The authors declare that they have no conflict of interest.

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
