## [Reviewer comments · Life Science Alliance]

Life Science Alliance

Mechanical forces and ligand-binding modulate *Pseudomonas aeruginosa* PilY1 mechanosensitive protein

Francisco Cao-Garcia, Jane Walker, Stephanie Board, and Alvaro Alonso-Caballero

DOI: <https://doi.org/10.26508/lsa.202403111>

Corresponding author(s): Alvaro Alonso-Caballero, Universidad Autónoma de Madrid

Review Timeline:

Submission Date:	2024-10-24
Editorial Decision:	2024-12-02
Revision Received:	2025-02-07
Editorial Decision:	2025-02-24
Revision Received:	2025-02-28
Accepted:	2025-02-28

Transaction Report:

December 2, 2024

Re: Life Science Alliance manuscript #LSA-2024-03111-T

Alvaro Alonso-Caballero
Universidad Autonoma de Madrid
Spain

Dear Dr. Alonso-Caballero,

Thank you for submitting your manuscript entitled "Mechanical forces and ligand-binding modulate *Pseudomonas aeruginosa* PilY1 mechanosensitive protein" to Life Science Alliance. The manuscript was assessed by expert reviewers, whose comments are appended to this letter. We invite you to submit a revised manuscript addressing the Reviewer comments.

Thank you for this interesting contribution to Life Science Alliance. We are looking forward to receiving your revised manuscript.

Sincerely,

B. MANUSCRIPT ORGANIZATION AND FORMATTING:

Reviewer #1 (Comments to the Authors (Required)):

Here the investigators report the impact of mechanical force and ligand binding on PilY1 protein behavior. They conclude from their studies that at higher forces (>20 pN), the Ca-binding domain plays a key role in maintaining protein structure. At lower forces (<7pN) they propose that the integrin-binding domain displays features of a mechanosensitive protein. They observe single integrin binding/unbinding events as a function of applied force. They propose that different forces act on different parts of the protein.

Overall, this is an interesting study. I have a number of suggestions below, including additional experiments, as well as questions about statistics. I am not an expert in the use of single-molecule magnetic tweezers, but the authors do a good job of explaining their finding to a non-expert. An expert in the technique will be able to comment on key technical aspects of the work.

Specific comments

1. This paper (PMID: 39477930) like came out after this manuscript was submitted. It would be useful to cite this paper and incorporate it into your discussion.
2. Adding line number would have been helpful for the review.
3. I am not sure of the use of the work "ligand" in the title and manuscript. I believe here the authors are referring to Ca? This is really more of a co-factor for the protein, as opposed to a free ligand that when bound (reversibly) changes the proteins activation state. I would rethink the use of the word "ligand" to not confuse the readers. Alternatively, given that Ca in this case does release, perhaps mention that Ca, and I assume also integrins, are the ligands you are referring to. Just be clear about these points in the abstract.
4. I would argue that the AFM data in reference #18 shows a link between the biophysical properties of the protein and its ability to sense surface. The other references cited in this section, at best, establish genetic associations.
5. "Our results indicate that PilY1 is a mechanosensor protein that exhibits different behavior depending on the force load." This is a terrific summary of the work and should also be added to the abstract. I would even say that the last paragraph of the introduction is a better abstract than the abstract!
6. A key strength of this manuscript is the technique used, which has not been applied to pure PilY1 protein in this way.
7. One concern is that they are using a tagged protein. Is there any evidence that the tag does (or does not) interfere with protein function? This point should at least be mentioned.
8. Fig. 1B - As written, this is confusing because you are citing a PDB file and alpha fold.
9. Figure 1D. I do not see statistics to determine if the EGTA-mediated changes are statistically significant.
10. Fig, 1D. While EGTA has reasonable specificity for Ca, I think at least one experiment adding back Ca versus another divalent cation to EGTA-treated samples would be important to confirm Ca specificity.
11. One concern (perhaps I have missed this) but I do not see statistical analyses associated with any of the data.
12. "To prove this hypothesis" should really be "to test this hypothesis".
13. You show data for "low" and "high" force trajectories. What happens at the mid-range of 7-20 pN?
14. "The protein exhibits a mixed mechanical behavior, where the unfolding force distributions of I1 and I2 are like the ones observed with EGTA, but the variable and last intermediates reproduce the Ca²⁺ condition results". This gets to point #10 above, that perhaps it is NOT all about Ca.
15. I would imagine some control protein beside integrin should be included in these studies?
16. "How the Ca²⁺-bound state of PilY1 antagonizes T4P retraction is not yet clear" - could it be that Ca-binding enhances binding of the pilus to the surface, thus making retraction more difficult? That is - PilY1 is a better adhesin plus Ca.

Reviewer #2 (Comments to the Authors (Required)):

Summary:

This paper investigates the structural dynamics of the *Pseudomonas aeruginosa* Type IV Pilus tip protein PilY1 and its role in adhesion and mechanosensing. The researchers examined how specific regions of the PilY1 protein react to mechanical force and integrin ligand-binding in order to test its mechanosensing capabilities. Truncated PilY1501-1161 is used in order to study the C-terminal domain alone. Through the use of largely one technique, single molecule magnetic tweezing, PilY1's unfolding and refolding behaviours under different force conditions are examined. Distinct unfolding intermediates are found under

increasing mechanical force. I1, I2, variable intermediates and a last intermediate while Ca^{2+} stabilises specific structures and reduces variability, making the protein more resistant to force. PilY1501-1161 also alternates between a folded and unfolded state at constant low force which is indicative of a mechanosensor protein. Next, the integrin binding domain was removed for the construct so that only the β -propeller region remained (PilY1642-1161). This led to the elimination of the I1 and I2 intermediates which were the regions suggested in the protein's mechanosensitivity and adhesion function. Calcium ion conditions still led to stabilisation. Integrin binding was also tested on PilY1501-1161 and it led to stabilisation of I2 under force, preventing full unfolding and maintaining in a partially unfolded state. This stabilisation is consistent with a protein's ability for adhesion to host cells, as integrin binding reinforces its structure under mechanical load, supporting bacterial attachment.

Positives:

The study provides an in-depth look at the structural dynamics of the PilY1 C-terminal domains. By using single-molecule magnetic tweezers, the researchers were able to test the strength of PilY1 folding at different conditions (e.g. +/- Ca^{2+} , integrin binding). The manuscript is well written with detailed figures that were relatively easy to follow.

The following points should be addressed or better clarified.

1. There were no tests done on the full-length PilY1 protein. While the intact PilY1 might be difficult to produce recombinantly, the authors should consider examining the N-terminal VWA domain. A very recent study (Guo et al 2024, Nat Comm) indicated that the N-terminal VWA domain is at the exterior of PilY1, not the C-terminal domains (IBD+ beta propellor). Thus, it is possible that VWA domain is poised to interact with surface or ligand before the integrin-binding domain. It would thus be crucial for the authors to test a construct that spans the VWA domain along, or with the VWA and the integrin-binding domain together (e.g. PilY1 1-642), if this can be produced recombinantly.

In Figure 1B, the authors show an alphafold model of the PilY1 C-terminal domains. The full-length PilY1 model would indicate that the integrin-binding domain is localised in between the VWA and beta-propellor domains, and it likely interacts closely with the VWA domain. The IBD may be partially occluded by the VWA, altering its accessibility to ligands like integrin. Again, it would be more informative, the experiment can be done with a VWA-spanning construct. In addition, the VWF domain is commonly known for its role in adhesion in bacteria, and is present in integrin receptors in mammalian cells. Therefore, studying this VWA domain would be key to support the authors' claims in PilY1's roles as an adhesin and mechanosensory.

The integrin-binding experiments are intriguing, but they lack specificity tests with other ligands or controls to confirm that the observed effects are uniquely attributable to integrin interactions. For instance, in figure 5, as a control, why not also test the effects of integrin addition on PilY1642-1161. This additional control experiment with PilY1642-1161 and integrin added would give more concrete evidence as to the β -propeller's role in integrin binding.

As discussed, the authors employed the single-molecule tweezing as the main technique for this study. It would be good to supply some biochemical data. For instance, even SDS-PAGE and chromatographic profiles to give indications of the integrity and purity of PilY1 proteins will be welcomed. Can the authors also supply biophysical data recapitulating complex formation of the integrin and PilY1? This will complement the tweezing results.

The authors are also welcome to discuss limitations of the single-molecule approach, such as potential artifacts introduced by the tethering constructs, for researchers in the field not familiar with this biophysical method.

Minor points.

Some text and graphs in the figures were very small when printed. The authors can consider making certain graphs two-column width to increase their legibility. For instance, Fig 1a, 2a, 4b can be two column width to help see the insets better.

The authors could speculate more on how the interplay between mechanical forces and ligand-binding regulates PilY1's functions across different environmental conditions.

Reviewer #1 (Comments to the Authors (Required)):

Here the investigators report the impact of mechanical force and ligand binding on PilY1 protein behavior. They conclude from their studies that at higher forces (>20 pN), the Ca-binding domain plays a key role in maintaining protein structure. At lower forces (<7pN) they propose that the integrin-binding domain displays features of a mechanosensitive protein. They observe single integrin binding/unbinding events as a function of applied force. They propose that different forces act on different parts of the protein.

Overall, this is an interesting study. I have a number of suggestions below, including additional experiments, as well as questions about statistics. I am not an expert in the use of single-molecule magnetic tweezers, but the authors do a good job of explaining their finding to a non-expert. An expert in the technique will be able to comment on key technical aspects of the work.

We acknowledge the referee for the recognition granted to our results and manuscript presentation, for the detailed reading of the manuscript, and the constructive recommendations. We have carefully addressed all the recommendations as detailed below.

Specific comments

1. This paper (PMID: 39477930) like came out after this manuscript was submitted. It would be useful to cite this paper and incorporate it into your discussion.

We have referenced and included this paper in the discussion:

Lines 466-468: *This would ensure the structural integrity of the pilus-tip complex for priming pilus polymerization in a subsequent extension cycle (Treuner-Lange et al, 2020; Guo et al, 2024).*

Lines 516-517: *Future work should address the nanomechanics of the vWA domain, given its central role in bacterial mechanotransduction and its spatial location in PilY1. Structural models indicate that the vWA domain may be blocking the integrin-binding site, meaning this site might only become accessible for ligand binding after the domain undergoes force-induced conformational changes (Guo et al, 2024).*

2. Adding line number would have been helpful for the review.

We apologize for missing this detail. We have included line numbers in the manuscript and supplementary information.

3. I am not sure of the use of the work "ligand" in the title and manuscript. I believe here the authors are referring to Ca? This is really more of a co-factor for the protein, as opposed to a free ligand that when bound (reversibly) changes the proteins activation state. I would rethink the use of the word "ligand" to not confuse the readers. Alternatively, given that Ca in this case

does release, perhaps mention that Ca, and I assume also integrins, are the ligands you are referring to. Just be clear about these points in the abstract.

We have revised the abstract to clearly state that we refer to calcium and integrin as ligands.

4. I would argue that the AFM data in reference #18 shows a link between the biophysical properties of the protein and its ability to sense surface. The other references cited in this section, at best, establish genetic associations.

We have clarified that the PilY1 N-terminal domain role in surface sensing has been inferred with genetics, surface mechanics (Siryaporn et al, 2014; Wang et al, 2023), and biophysical studies (Webster et al., 2022).

Lines 72-75: While genetic, surface mechanics (Siryaporn et al, 2014; Wang et al, 2023), and biophysical (Webster et al, 2022) studies support the vWA domain's involvement in mechanosensing, less is known about how force could regulate the functions of the C-terminal section of PilY1.

5. "Our results indicate that PilY1 is a mechanosensor protein that exhibits different behavior depending on the force load." This is a terrific summary of the work and should also be added to the abstract. I would even say that the last paragraph of the introduction is a better abstract than the abstract!

We appreciate the reviewer's suggestion, which helped us improve the abstract. Now this is the central sentence of the revised abstract (as noted in point #3) and the summary blurb.

6. A key strength of this manuscript is the technique used, which has not been applied to pure PilY1 protein in this way.

We appreciate the reviewer for highlighting this. We have clearly emphasized the technique in the main sentence of the abstract and the summary blurb (see previous item).

7. One concern is that they are using a tagged protein. Is there any evidence that the tag does (or does not) interfere with protein function? This point should at least be mentioned.

Both reviewers have raised this question (Reviewer #1 in point #7 and Reviewer #2 in point #3b):

HaloTag and I32 titin domain tags exhibit high mechanical stability (Li et al., 2002; Popa et al., 2013). At the forces explored in this study, the unfolding probability of any of these domains is low; thus, they remain folded and function as rigid handles during the experiments. In the rare event of unfolding, the tag can be easily identified based on the extension and its known sequence and structure. In the trajectory shown in Supplementary Figure 15, recorded at 5.5 pN, the unfolding of one I32 domain (resulting in an extension increase of 13 nm, indicated by an arrow) shifts the protein's offset extension and increases the signal's noise (increased

fluctuation after the release of additional unstructured polypeptide). Nevertheless, the dynamics of I1 and I2 continue unaltered. The grey dotted lines indicate the unfolded level of I1 and the folded level of I2 for reference. We have clarified this point in the Results (lines 130-133) and Material and Methods (lines 751-753) sections, and we have added Supplementary Figure 15.

Lines 130-133: *HaloTag and I32 domains exhibit high mechanical stability and act as rigid molecular handles (Li et al, 2002; Popa et al, 2013). At the forces tested in this study, their probability of unfolding is low, and they do not exhibit conformational changes during the experiments (see Material and Methods).*

Lines 751-753: *At the forces tested, the unfolding probability of the HaloTag or I32 domains remains low. In the rare event of unfolding, their identity can be easily ascertained and does not alter PilY1 behavior (Supplementary Fig. 15).*

8. Fig. 1B - As written, this is confusing because you are citing a PDB file and alpha fold.

We have corrected this sentence and clarified the references.

Lines 548-549: *B) Crystal structure of PilY1501-1161 from top (left) and side (right) views (prediction from AlphaFold (Jumper et al, 2021; Varadi et al, 2022) and color assignation based on reference (Orans et al, 2010)).*

9. Figure 1D. I do not see statistics to determine if the EGTA-mediated changes are statistically significant.

The statistical analysis used is indicated in the Material and Methods section (lines 765-770), and the results are shown in the Supplementary Information in Tables S1-S3. To assess the significance of the difference between the unfolding forces between buffer conditions (EGTA or Ca^{2+}) and between intermediates, the difference between the means and its uncertainty is computed (from the SEM of the values at each condition). The difference is stated significant when the probability of being zero is smaller than 5% ($p < 0.05$) (Trosset, 2009).

Lines 765-770: *To assess the significance of the difference between the unfolding forces between buffer conditions (EGTA or Ca^{2+}) and between intermediates (Supplementary Tables S1-3), the difference of the means and its uncertainty is computed (from the SEM of the values at each condition). The difference is stated significant when the probability of being zero is smaller than 5% ($t > t_{\text{critical}}$, being $t_{\text{critical}}=2$ for $p < 0.05$) (Trosset, 2009).*

10. Fig, 1D. While EGTA has reasonable specificity for Ca, I think at least one experiment adding back Ca versus another divalent cation to EGTA-treated samples would be important to confirm Ca specificity.

We have carried out the experiment proposed by the reviewer to test the calcium-binding specificity (Supplementary Fig. 2). On a single protein, we conduct a series of force-ramp

extensions under different buffer compositions (at least 10 repetitions per buffer), which we iteratively change from (1) magnesium to (2) calcium, (3) EGTA, (4) manganese, and (5) calcium again. Supplementary Fig. 2A shows three consecutive example ramps per condition over the duration of the experiment, with breaks in time continuity indicated by symbols placed between the extension and force axes. The force is quenched between ramps to allow protein folding and repeat the unfolding ramp protocol. We have analyzed the last unfolding force values under the different buffer conditions (Supplementary Fig. 2A and 2B), and the results show that forces do not change significantly (computed difference of the unfolding force means and their uncertainty, Supplementary Fig. 2C) between conditions, except for EGTA and calcium (both before and after EGTA treatment). Magnesium and manganese shift the unfolding force to higher values, but the differences are not significant when both treatments are compared to EGTA or calcium. Previous studies explored the affinity of different cations toward PilY1 calcium-binding sites (Orans et al., 2010; Johnson et al., 2011). Our experiment is constrained by sample size and only reflects the variability of a single molecular entity. Understanding how different cations affect PilY1 mechanical stability would necessitate a more comprehensive study testing multiple molecules, as we did by comparing EGTA and calcium. Nevertheless, in this experiment, we have observed how various buffer compositions may influence one PilY1 molecule's mechanical stability, highlighting the specific effect of calcium on its nanomechanics.

Lines 206-208: *Ca²⁺ specifically induces this effect on the mechanical stability of the protein, though other divalent cations can also alter PilY1's nanomechanics (Supplementary Fig. 2).*

11. One concern (perhaps I have missed this) but I do not see statistical analyses associated with any of the data.

We have clearly outlined the statistical methods employed to determine statistical significance in the Materials and Methods section, as well as in the captions of Supplementary Tables S1-S3. Refer to point #9.

12. "To prove this hypothesis" should really be "to test this hypothesis".

We have corrected this sentence, as suggested (line 359).

13. You show data for "low" and "high" force trajectories. What happens at the mid-range of 7-20 pN?

Above 7 pN, the unfolding of I1 and I2 occurs in shorter times, as expected, with an increasing applied force. Subsequent intermediates (I3 to Last intermediate) exhibit higher mechanical stability than I1 and I2, and the timescales to observe their unfolding below 20 pN are impractical from the experimental perspective. Supplementary Figure 5 shows that at an intermediate force (12 pN), the unfolding of I1 and I2 occurs rapidly, whereas the subsequent intermediate (I3) takes approximately 300 seconds to unfold. We have clarified this point in the text (lines 276-280 and Supplementary Fig. 5), where we emphasize the presence of two

ranges of mechanostability affecting I1 and I2 on one side and the variable and final intermediates on the other.

Lines 276-280: *The sequential unfolding of PilY1501-1161 suggests that I1 and I2 are less stable than the other intermediates, which divides the protein into two ranges of mechanostability (Supplementary Fig. 5).*

14. "The protein exhibits a mixed mechanical behavior, where the unfolding force distributions of I1 and I2 are like the ones observed with EGTA, but the variable and last intermediates reproduce the Ca²⁺ condition results". This gets to point #10 above, that perhaps it is NOT all about Ca.

We have addressed the reviewer's point by conducting the experiment shown in Supplementary Fig 2. See point #10.

15. I would imagine some control protein beside integrin should be included in these studies?

Reviewer #2 suggested a similar test, which is also included in the answer to Reviewer's #2 point #2:

We have included Supplementary Figure 11, which shows control experiments with 75 μ M BSA, a concentration two orders of magnitude higher than that used to test integrin binding (200 nM). At three reference forces (Supplementary Fig. 11A), the extension changes of I1 and I2 remain constant and overlap with the non-BSA condition (Supplementary Fig. 11B), demonstrating that the shortening observed in PilY1⁵⁰¹⁻¹¹⁶¹ (Fig. 5) is unequivocally due to integrin binding (lines 380-383).

Lines 380-383: *This shortening of the I2 unfolded extension only occurs and is specifically induced in the presence of integrin (Supplementary Fig. 11).*

16. "How the Ca²⁺-bound state of PilY1 antagonizes T4P retraction is not yet clear" - could it be that Ca-binding enhances binding of the pilus to the surface, thus making retraction more difficult? That is - PilY1 is a better adhesin plus Ca.

It has been reported that calcium binding enhances the mechanical stability of various pilin proteins (Echelmann et al., 2016; Oude Vrielink et al., 2017; Milles et al., 2018). This increase in mechanical stability would ensure the structural integrity of the pili adhering to the surface, thus preventing the detachment of the bacterium. We have included this concept in the discussion section (lines 468-472), along with the "cork" hypothesis proposed by Treuner-Lang et al., 2020 and Guo et al 2024.

Lines 468-472: *It is also feasible that the increased mechanical stabilization of PilY1 by Ca²⁺ could be linked to its adhesive role, contributing to the structural integrity of this protein during surface attachment, as has been proposed for other bacterial pilin proteins (Echelmann et al, 2016; Oude Vrielink et al, 2017; Milles et al, 2018).*

Reviewer #2 (Comments to the Authors (Required)):

Summary:

This paper investigates the structural dynamics of the *Pseudomonas aeruginosa* Type IV Pilus tip protein PilY1 and its role in adhesion and mechanosensing. The researchers examined how specific regions of the PilY1 protein react to mechanical force and integrin ligand-binding in order to test its mechanosensing capabilities. Truncated PilY1501-1161 is used in order to study the C-terminal domain alone. Through the use of largely one technique, single molecule magnetic tweezing, PilY1's unfolding and refolding behaviours under different force conditions are examined. Distinct unfolding intermediates are found under increasing mechanical force. I1, I2, variable intermediates and a last intermediate while Ca^{2+} stabilises specific structures and reduces variability, making the protein more resistant to force. PilY1501-1161 also alternates between a folded and unfolded state at constant low force which is indicative of a mechanosensor protein. Next, the integrin binding domain was removed for the construct so that only the β -propeller region remained (PilY1642-1161). This led to the elimination of the I1 and I2 intermediates which were the regions suggested in the proteins mechanosensitivity and adhesion function. Calcium ion conditions still led to stabilisation. Integrin binding was also tested on PilY1501-1161 and it led to stabilisation of I2 under force, preventing full unfolding and maintaining in a partially unfolded state. This stabilisation is consistent with a protein's ability for adhesion to host cells, as integrin binding reinforces its structure under mechanical load, supporting bacterial attachment.

Positives:

The study provides an in-depth look at the structural dynamics of the PilY1 C-terminal domains. By using single-molecule magnetic tweezers, the researchers were able to test the strength of PilY1 folding at different conditions (e.g. +/- Ca^{2+} , integrin binding). The manuscript is well written with detailed figures that were relatively easy to follow.

We acknowledge the referee for the recognition granted to our results and manuscript presentation, for the detailed reading of the manuscript, and the constructive recommendations. We have carefully addressed all the recommendations as detailed below.

The following points should be addressed or better clarified.

1. There were no tests done on the full-length PilY1 protein. While the intact PilY1 might be difficult to produce recombinantly, the authors should consider examine the N-terminal VWA domain. A very recent study (Guo et al 2024, Nat Comm) indicated that the N-terminal VWA domain is at the exterior of PilY1, not the C-terminal domains (IBD+ beta propellor). Thus, it is possible that VWA domain is posed to interact with surface or ligand before the integrin-binding domain. It would thus be crucial for the authors to test a construct that span the VWA domain along, or with the VWA and the integrin-binding domain together (e.g. PilY1 1-642), if this can be produced recombinantly.

In Figure 1B, the authors show an alpha-fold model of the PilY1 C-terminal domains. The full-length PilY1 model would indicate that integrin-binding domain is localised in between the VWA and beta-propellor domains, and it likely interacts closely with the VWA domain. The IBD may be partially occluded by the VWA, altering its accessibility to ligands like integrin. Again, it would be more informative, the experiment can be done with a VWA-spanning construct. In addition, the VWF domain is commonly known for its role in adhesion in bacteria, and is present in integrin receptors in mammalian cells. Therefore, studying this VWA domain would be key to support the authors claims in PilY1's roles as an adhesin and mechanosensory.

We understand the reviewer's point. Initially, we aimed to dissect the nanomechanics of PilY1 by exploring its N-terminal and C-terminal sections. We designed an N-terminal construct that includes the vWA domain. However, we encountered challenges in producing, purifying, and measuring this portion of PilY1, which, to our knowledge, has remained elusive to structural determination (Hernández-Sánchez et al, 2024). We carried out limited measurements where we observed low conformational reproducibility and complex dynamics, and we are unsure if this behavior corresponds to the actual properties of the vWA domain. If we resolve these experimental challenges in the future, we will explore this section of the protein in depth, which, as the reviewer indicates, would underpin the mechanosensing role of PilY1. Nevertheless, our single-molecule approach to PilY1's C-terminal region dynamics provides key insights into how mechanical forces and ligand binding could influence this protein's roles as an adhesin and a T4P regulator.

2. The integrin-binding experiments are intriguing, but they lack specificity tests with other ligands or controls to confirm that the observed effects are uniquely attributable to integrin interactions. For instance, in figure 5, as a control, why not also test the effects of integrin addition on PilY1642-1161. This additional control experiment with PilY1642-1161 and integrin added would give more concrete evidence as to the β -propeller's role in integrin binding.

Reviewer #1 suggested a similar test, which we replied to in Reviewer's #1 point #15:

We have included Supplementary Figure 11, which shows control experiments with 75 μ M BSA, a concentration two orders of magnitude higher than that used to test integrin binding (200 nM). At three reference forces (Supplementary Fig. 11A), the extension changes of I1 and I2 remain constant and overlap with the non-BSA condition (Supplementary Fig. 11B), demonstrating that the shortening observed in PilY1⁵⁰¹⁻¹¹⁶¹ (Fig. 5) is unequivocally due to integrin binding (lines 380-383).

Lines 380-383: *This shortening of the I2 unfolded extension only occurs and is specifically induced in the presence of integrin (Supplementary Fig. 11).*

Additionally, Reviewer #2 suggests an alternative control experiment (Supplementary Figure 12) using the construct PilY1⁶⁴²⁻¹¹⁶¹ to determine if integrin binds to this region of PilY1 without the integrin binding site (located in residues 617-619). PilY1⁶⁴²⁻¹¹⁶¹ intermediates (from I3 to the last intermediate) do not show conformational changes at the forces where we detected integrin binding in the PilY1⁵⁰¹⁻¹¹⁶¹ construct (below 7 pN; refer to the response to Reviewer #1 in point #13, which addresses the two mechanostability ranges of the protein). Therefore, no

change in the protein's extension can be observed as no structures undergo unfolding. We designed an experiment to evaluate integrin binding to PilY1⁶⁴²⁻¹¹⁶¹, repeating a protocol of unfolding at 40 pN, followed by quenching to 5.0 pN (for varying time) to facilitate folding and the potential binding of integrin, and then reassessing the unfolding trajectory at 40 pN. We compared the final extension after unfolding at 40 pN in the absence or presence of 800 nM of integrin (four times the concentration tested on the PilY1⁵⁰¹⁻¹¹⁶¹ construct). If integrin binds to PilY1⁶⁴²⁻¹¹⁶¹, we could potentially see a change in the final unfolded extension. No changes were detected, suggesting that integrin binding is specific and requires the presence of the RGD motif (residues 617-619) for binding and inducing a change in the unfolded extension of PilY1. The requirement for the RGD motif in integrin binding to PilY1 was demonstrated previously (Johnson et al., 2011). In our case, the PilY1⁶⁴²⁻¹¹⁶¹ construct lacks the 141-residue sequence present in the PilY1⁵⁰¹⁻¹¹⁶¹ construct, including the RGD motif and the essential calcium-binding site required for integrin binding. Our results reinforce this previous finding that integrin association with PilY1 depends on the presence of the RGD motif (lines 393-397).

Lines 393-397: Integrin recruitment occurs only in the presence of the RGD motif (residues 617-619) present in the PilY1501-1161 construct (Johnson et al, 2011), which bestows the specificity of the PilY1-integrin interaction (Supplementary Fig. 12).

3. As discussed, the authors employed the single-molecule tweezing as the main technique for this study. It would be good to supply some biochemical data. For instance, even SDS-PAGE and chromatographic profiles to give indications of the integrity and purity of PilY1 proteins will be welcomed. Can the authors also supply biophysical data recapitulating complex formation of the integrin and PilY1? This will complement the tweezing results.

In Supplementary Figure 14, we have included the FPLC size-exclusion chromatographs for both PilY1 constructs, along with an SDS-PAGE gel displaying the peak fractions highlighted in the chromatographs to demonstrate the purity of the proteins (lines 684-685). While we lack access to additional biophysical tools to monitor complex formation, integrin-PilY1 binding was previously demonstrated, as noted in point #2 (Johnson et al., 2011). Our single-molecule observations, where we can detect integrin-binding events in real-time as a conformational change in PilY1, underpin and support the previously described interaction.

Lines 684-685: Protein purification was assessed using SDS-PAGE. (Supplementary Fig. 14).

The authors are also welcome to discuss limitations of the single-molecule approach, such as potential artifacts introduced by the tethering constructs, for researchers in the field not familiar with this biophysical method.

Both reviewers have raised this question (Reviewer #1 in point #7 and Reviewer #2 in point #3b)

HaloTag and I32 titin domain tags exhibit high mechanical stability (Li et al., 2002; Popa et al., 2013). At the forces explored in this study, the unfolding probability of any of these domains is low; thus, they remain folded and function as rigid handles during the experiments. In the rare

event of unfolding, the tag can be easily identified based on the extension and its known sequence and structure. In the trajectory shown in Supplementary Figure 15, recorded at 5.5 pN, the unfolding of one I32 domain (resulting in an extension increase of 13 nm, indicated by an arrow) shifts the protein's offset extension and increases the signal's noise (increased fluctuation after the release of additional unstructured polypeptide). Nevertheless, the dynamics of I1 and I2 continue unaltered. The grey dotted lines indicate the unfolded level of I1 and the folded level of I2 for reference. We have clarified this point in the Results (lines 130-133) and Material and Methods (lines 751-753) sections, and we have added Supplementary Figure 15.

Lines 130-133: *HaloTag and I32 domains exhibit high mechanical stability and act as rigid molecular handles (Li et al, 2002; Popa et al, 2013). At the forces tested in this study, their probability of unfolding is low, and they do not exhibit conformational changes during the experiments (see Material and Methods).*

Lines 751-753: *At the forces tested, the unfolding probability of the HaloTag or I32 domains remains low. In the rare event of unfolding, their identity can be easily ascertained and does not alter PilY1 behavior (Supplementary Fig. 15).*

Minor points.

Some text and graphs in the figures were very small when printed. The authors can consider making certain graphs two-column width to increase their legibility. For instance, Fig1a, 2a, 4b can be two column width to help see the insets better.

We have implemented the changes recommended by the reviewer. We understand that the reviewer is referring to Fig. 1C instead of 1A.

The authors could speculate more on how the interplay between mechanical forces and ligand-binding regulates PilY1's functions across different environmental conditions.

We have expanded the discussion (lines 474-480) on how PilY1 force and ligand binding activities may be modulated in various environments or in relation to specific hosts, for which we have included references (Cruz et al., 2012, 2014; Parker et al., 2015; Herfurth et al., 2022; Hernández-Sánchez et al., 2024).

Lines 474-480: *This would entail that the Ca²⁺-mediated binding and T4P regulation could be finely tuned to the chemical environment of bacteria (Morand et al, 2004; Cruz et al, 2014; Parker et al, 2015). Similar to chemical affinity, the nanomechanics of PilY1 are probably adapted to the environment and are specific to the host. This specificity would create a unique interplay between mechanical forces and ligand binding in regulating PilY1 functions, depending on the substrate being colonized. (Cruz et al, 2012, 2014; Parker et al, 2015; Herfurth et al, 2022; Hernández-Sánchez et al, 2024).*

We sincerely appreciate the thorough reading of the paper by both reviewers, along with their constructive feedback. We have addressed all their suggestions, which have helped enhance

the manuscript. We expect the present version of the manuscript is ready to be published in Life Science Alliance.

February 24, 2025

RE: Life Science Alliance Manuscript #LSA-2024-03111-TR

Dr. Alvaro Alonso-Caballero
Universidad Autónoma de Madrid
C/Darwin, 2
Madrid 28049
Spain

Dear Dr. Alonso-Caballero,

Thank you for submitting your revised manuscript entitled "Mechanical forces and ligand-binding modulate *Pseudomonas aeruginosa* PilY1 mechanosensitive protein". We would be happy to publish your paper in Life Science Alliance pending final revisions necessary to meet our formatting guidelines.

- please be sure that the authorship listing and order is correct
- please add the Twitter/X and Bluesky handles of your host institute/organization as well as your own or/and one of the authors in our system
- please note that supplementary figures should be provided only separately
- please remove figures from the manuscript file...they should be uploaded only separately
- please consult our manuscript preparation guidelines <https://www.life-science-alliance.org/manuscript-prep> and make sure your manuscript sections are in the correct order
- please add your main, supplementary figure, and table legends to the main manuscript text after the references section
- please label the panel C in Figure S1
- please add callouts for Figures S2A-C; S6D; S7D; S8A-C; S9A-C; S10A-G and S11A-B to your main manuscript text

FIGURE CHECK:

- please add sizes next to the blots in Figure S14

A. FINAL FILES:

B. MANUSCRIPT ORGANIZATION AND FORMATTING:

Sincerely,

Reviewer #1 (Comments to the Authors (Required)):

The authors have addressed all my concerns.

February 28, 2025

RE: Life Science Alliance Manuscript #LSA-2024-03111-TRR

Dr. Alvaro Alonso-Caballero
Universidad Autónoma de Madrid
C/Darwin, 2
Madrid 28049
Spain

Dear Dr. Alonso-Caballero,

Thank you for submitting your Research Article entitled "Mechanical forces and ligand-binding modulate *Pseudomonas aeruginosa* PilY1 mechanosensitive protein". It is a pleasure to let you know that your manuscript is now accepted for publication in Life Science Alliance. Congratulations on this interesting work.

DISTRIBUTION OF MATERIALS:

Again, congratulations on a very nice paper. I hope you found the review process to be constructive and are pleased with how the manuscript was handled editorially. We look forward to future exciting submissions from your lab.

Sincerely,
